# Muscle fiber Myc is dispensable for muscle growth and its forced expression severely perturbs homeostasis

Daniel J. Ham [1] ✉, Michelangelo Semeraro[1], Bianca M. Berger [1,2], Timothy J. McGowan [1], Shuo Lin[1], Eleonora Maino [1], Filippo Oliveri[1] & Markus A. Rüegg [1] ✉

The oncogenic transcription factor Myc stimulates many growth processes including cell cycle progression and ribosome biogenesis. Myc expression is low in adult skeletal muscle, but is upregulated upon growth stimuli. Furthermore, muscle fiber Myc overexpression recapitulates many aspects of growth-related gene expression, leading to the hypothesis that Myc mediates pro-growth responses to anabolic stimuli, such as exercise. Here, we test this hypothesis by examining mouse models in which Myc is specifically eliminated or overexpressed in skeletal muscle fibers or muscle stem cells (MuSC). While muscle fiber Myc expression increased during muscle growth and Myc expression in MuSCs was required for successful muscle regeneration, muscle fiber Myc expression was dispensable for post-natal, mechanical overload or PKBα/Akt1-induced muscle growth in mice. Similarly, constitutive Myc expression did not promote skeletal muscle hypertrophy, but instead impaired muscle fiber structure and function within days. These data question the role of Myc in skeletal muscle growth.

Skeletal muscle has a remarkable capacity to adapt to chronic changes in its use by changing the size, number and composition of the fibers whose contractility allows force generation. In response to increased mechanical load, such as resistance training or reloading after a prolonged period of inactivity, skeletal muscle grows. To support growth, many new muscle proteins must be synthesized. Rapid, acute increases in protein synthesis are largely mediated via post-translational modifications (i.e. phosphorylation) of proteins, particularly by the protein kinase B (PKB)/Akt-mammalian target of rapamycin complex 1 (mTORC1) pathway. However, in response to sustained anabolic stimuli, increased translational capacity (i.e. ribosome biogenesis) is considered central to the success of this response, and coincides with expression of the powerful transcription factor c-Myc, hereafter referred to as Myc, which strongly promotes expression of ribosomal protein genes[1]. After forming a heterodimer with its obligate partner Max, Myc binds consensus E-boxes (CACGTG) in target genes, which are generally believed to comprise 10–15% of the genome and include many genes involved in cell cycle regulation, ribosome biogenesis, protein translation and metabolism[2].

In response to loading, a robust induction of *Myc* gene and/or protein expression has been observed in both rodents[3–11] and healthy young humans[12–19]. To some extent, an acute pulse of Myc expression also recapitulates aspects of the load-induced gene expression response, especially those related to ribosome biogenesis[8,20]. Longer term (2 weeks) transgenic Myc expression has also been shown to replicate exercise-induced increases in protein synthesis, although the response was uncoupled from muscle growth[21]. Furthermore, along with a blunted hypertrophic response, the exercise-induced increase in *MYC* transcripts and/or protein abundance is blunted in elderly human muscle[13,20,22–24]. Recently, premature aging phenotypes and gene expression signatures were observed in skeletal muscle tissue, among

[1]Biozentrum, University of Basel, Basel, Switzerland. [2]Present address: Institute of Cell Biology, University of Bern, Bern, Switzerland. ✉ e-mail: dan.ham@unibas.ch; markus-a.ruegg@unibas.ch

other tissues, when *Myc* was ablated at weaning in all tissues using the Cre/loxP system[25].

Based on the weight of this evidence, it has been proposed that Myc is at the center of the muscles' loading-responsive, pro-growth gene regulatory network and Myc's failed induction in aged muscle could be responsible for its subdued response to exercise[8,20]. However, although gene expression signatures consistent with muscle growth are induced in Myc transgenic mice, the necessity of muscle fiber *Myc* expression for skeletal muscle growth has not been directly tested. Furthermore, the robust skeletal muscle growth that occurs during early development or in response to mechanical overload also relies on the activity of non-muscle fiber cells, in particular skeletal muscle stem cells (MuSCs), which have been shown to rely heavily on Myc for their capacity to proliferate, differentiate and support muscle regeneration[26,27]. We, therefore, set out to characterize the role of muscle fiber and MuSC Myc in skeletal muscle growth and homeostasis.

To this end, we used muscle fiber (human skeletal actin; HSA) and MuSC (paired box 7; Pax7) specific promoters to drive Cre in combination with floxed alleles that either deplete (Myc floxed) or overexpress (MycTG) Myc. We conclusively show that muscle fiber Myc expression is not required for post-natal-, mechanical overload- (MOV) or PKBα/Akt1-induced skeletal muscle growth. On the other hand, MOV-induced muscle growth was almost completely abrogated by *Myc* depletion in MuSCs, which strongly impaired MuSC proliferation and the contribution of MuSCs to reloading-induced muscle growth following sciatic nerve injury. Furthermore, short-term transgenic *Myc* overexpression in MuSCs or muscle fibers was catastrophic for muscle homeostasis. These findings show that *Myc* expression needs to be tightly controlled and challenge the current view of Myc's role in skeletal muscle growth and homeostasis.

## Results

### Muscle fiber Myc expression is not necessary for post-natal muscle growth

To establish the role of muscle fiber Myc during normal post-natal muscle growth, we crossed HSA-Cre mice[28] with Myc fl/fl[29] mice to create HSA^wt/wt; Myc^fl/fl (CON) and HSA^Cre/wt; Myc^fl/fl (HSA-MycKO) mice and examined muscle mass and function. At 2 months of age, HSA-MycKO mice contained ~60% lower *Myc* mRNA content than CON mice in *quadriceps femoris* (QUAD) muscle, as measured via digital PCR (dPCR; Fig. 1A). Despite the strong reduction in *Myc* mRNA, expression of the muscle-specific ribosomal protein gene *Rpl3*, which was recently shown to be strongly induced by muscle fiber Myc overexpression[8], was unaltered. Importantly, many cell-types other than skeletal muscle fibers express Myc within skeletal muscle, which could explain the incomplete reduction in *Myc* expression in whole-muscle lysates of HSA-MycKO mice. With this in mind, we used RNA-fluorescent in situ hybridization (RNA-FISH) to localize *Myc* transcripts in *gastrocnemius* (GAS) muscle cross-sections from CON and HSA-MycKO mice. To distinguish *Myc* mRNA expression originating from muscle fiber nuclei, we co-stained sections with antibodies against laminin β1γ1 (to outline muscle fibers) and included *Pax7* probes to identify MuSCs, which are located between the basement membrane and sarcolemma and could easily be confused for muscle fiber nuclei. *Myc* mRNA could be clearly visualized in a small population of *Pax7*-expressing MuSCs, along with other mononucleated cells residing between muscle fibers, in muscle sections from both CON and HSA-MycKO mice (Fig. 1B). On the other hand, *Myc* mRNA puncta could only be seen within laminin-outlined muscle fibers of CON, but not HSA-MycKO mice, indicating successful, muscle fiber-specific abrogation of *Myc* transcripts.

Despite the strong reductions in muscle fiber *Myc* expression in HSA-MycKO mice, there was no obvious difference between genotypes in body mass for 2-month-old males or body mass and composition for 5- to 7-month-old male and female mice (Fig. 1C, D). Similarly, the mass of *tibialis anterior* (TA), QUAD, *extensor digitorum longus* (EDL), *soleus* (SOL), *plantaris* (PLA), GAS and *triceps brachii* (TRI) were comparable between genotypes in 2-month-old males (Fig. S1A) as well as 5- to 7-month-old males (Fig. S1B) and females (Fig. S1C). As differences in body mass largely account for differences in muscle mass in healthy, young adult mice, we explored relationships between TA, TRI, GAS and QUAD mass and body mass in male and female mice between 6 and 37 weeks of age for CON and HSA-MycKO mice (Fig. 1E). The mass of each muscle strongly correlated with body mass in both CON and HSA-MycKO mice, with the only differences in slope (TRI, males) or intercept (GAS) between genotypes indicating, if anything, a slightly higher muscle mass in HSA-MycKO than CON mice. Isolated muscle force recordings in *extensor digitorum longus* (EDL) muscles from 2-, and 5- to 7-month-old males showed identical responses of CON and HSA-MycKO muscles to stimulation frequencies from 50 to 250 Hz (Fig. 1F). On the other hand, peak EDL twitch force was slightly lower in male (Fig. 1F, G), but not female (Fig. S1D), HSA-MycKO than CON mice, without alteration in other twitch properties (Fig. 1G), while fatigability was comparable between genotypes. Similarly, specific force in the slow-type soleus muscle was comparable between genotypes (Fig. S1E). Together, these data indicate that muscle fiber MycKO has little impact on the development of normal muscle size and function.

Given Myc's well-established role as a powerful transcription factor targeting pro-growth gene networks[8], including all ribosomal protein genes, we performed mRNAseq on GAS muscle from CON and HSA-MycKO mice at 5 to 7 months of age. CON and HSA-MycKO samples clearly separated on principle component 1 (Fig. 1H) and a number of transcripts related to muscle structure (e.g. *Dmd*), the neuromuscular junction (NMJ; e.g. *Musk*) and TGFβ signaling (e.g. *Tgfbr1, Dkk3* and *Bmpr1b*) were all significantly lower in muscle from HSA-MycKO mice compared to CON mice (Fig. 1I). Interestingly, just 6 of the 239 downregulated (FDR < 0.05; Log fold <0) genes in HSA-MycKO muscle were in common with the 355 genes upregulated (FDR < 0.05; Log fold >0) in response to short-term muscle-specific Myc overexpression[8] (Fig. 1J). Furthermore, just one ribosomal protein gene (*Rpl13*) was significantly downregulated in HSA-MycKO mice, although there was a general trend for slightly lower ribosomal protein gene expression in HSA-MycKO mice (Fig. 1K). Importantly, the strong reduction in *Myc* expression did not coincide with compensatory upregulation of the other Myc-family paralogs *Mycl* and *Mycn*, which were expressed at levels ~10-fold lower than *Myc* (i.e. c-Myc) in 5- to 7-month-old CON mice (Fig. S1F). Together, these data indicate that muscle fiber Myc expression is not necessary for normal post-natal muscle growth, despite a downregulation of genes associated with muscle structure and development, potentially indicating that Myc-regulated growth processes are not sufficiently stressed during development for growth to be limited by its absence.

### Muscle fiber Myc expression is not necessary for mechanical overload-induced muscle growth

To test whether stronger growth stimuli would uncover a requirement for Myc, we next used mechanical overload (MOV) of the PLA muscle via ablation of the lower third of its synergist muscles (GAS and SOL), which is one of the most powerful experimental inducers of muscle growth. Myc is strongly increased in response to MOV, correlating with ribosome biogenesis[9] and the extent of muscle growth[17]. To examine whether muscle fiber Myc expression is necessary to support MOV-induced muscle growth, we performed synergist ablation on CON and HSA-MycKO mice (Fig. 2A) and examined muscle growth 7, 14 and 28 days later. After an initial inflammation-based increase in muscle mass 3 days after synergist ablation (Fig. 2B and S2A), muscle mass was not different between the ablated and contralateral leg at 7 days, before a 31% and 65% increase in muscle mass 14 and 28 days after ablation, respectively, in CON mice (Fig. 2B). Despite strong evidence to suggest that muscle fiber Myc expression contributes to MOV-

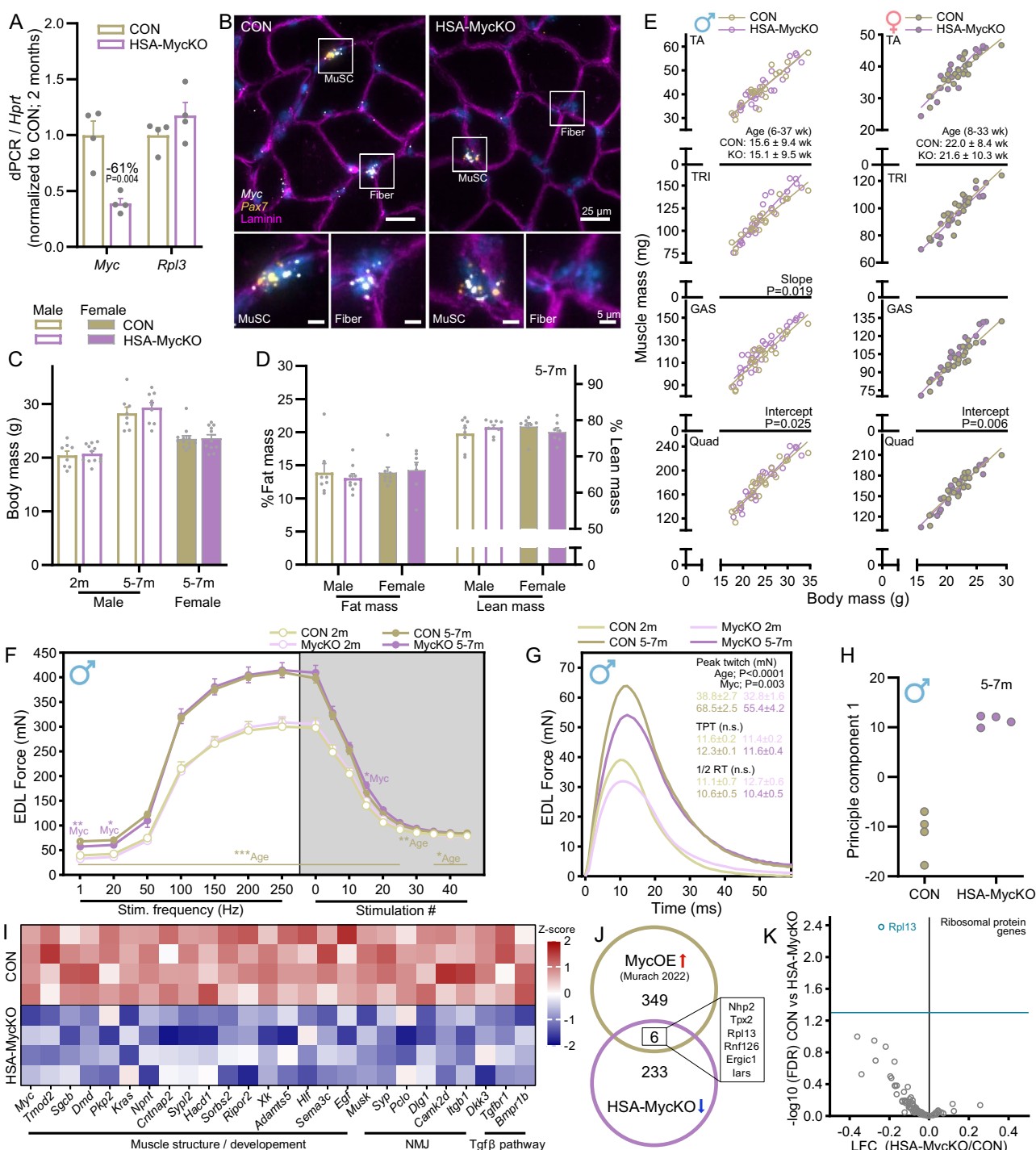

**Fig. 1 | Post-natal muscle growth is not altered in HSA-MycKO mice. A** Digital PCR for transcript levels of *Myc* and *Rpl3* normalized to a housekeeping gene (*Hprt*) in CON (HSA^wt/wt; Myc^fl/fl) and HSA-MycKO (HSA^Cre/WT; Myc^fl/fl) mice (*n* = 4). **B** Representative images of RNA fluorescent in situ hybridization (RNA-FISH) with probes against *Myc* (white) and *Pax7* (yellow; Muscle stem cells) and antibody staining for laminin β1γ1 (magenta). **C** Body mass for 2-month-old (2 m) males (CON, *n* = 8; KO, *n* = 12) and 5- to 7-month-old (5–7 m) males (CON, *n* = 8, KO, *n* = 9) and females (CON, *n* = 12; KO *n* = 11). **D** Whole-body measures of %fat (left) and % lean (right) mass (CON, *n* = 8; KO, *n* = 9 males, *n* = 8 females). **E** Correlations between body mass and TA, TRI, GAS and QUAD muscle mass for CON (*n* = 20–21 males, *n* = 19 females) and HSA-MycKO (*n* = 28 males, *n* = 20 females) mice. In vitro, EDL force frequency curve and fatigue response to multiple stimulations (**F**) and representative twitch force traces (**G**) for 2- (CON, *n* = 8; KO, *n* = 10–11) and 5- to 7-

month-old (CON, *n* = 4–5; KO, *n* = 6) male CON and HSA-MycKO mice. **H** Principle component analysis of mRNAseq data generated from the GAS muscle of 5- to 7-month-old male mice (*n* = 4) and (**I**) heatmap of select downregulated genes (FDR < 0.05) associated with muscle development, the neuromuscular junction (NMJ) and the TGFβ pathway. **J** Overlap of significantly downregulated genes in HSA-MycKO mice (FDR < 0.05) with a published list of genes upregulated by short-term muscle-specific Myc overexpression[8]. **K** Scatterplot of log-fold change and -log10 FDR for all ribosomal protein genes from mRNAseq data comparing CON and HSA-MycKO mice. Data are presented as mean ± SEM. Two-way ANOVA with Sidak's post hoc test (**A**, **C**, **D**, **F**, **G**) were used to compare between data. For (**I**–**K**), *P* values were adjusted for multiple comparisons using the Benjamini-Hochberg (FDR) procedure. *, **, and *** denote a significant difference between groups of *P* < 0.05, *P* < 0.01, and *P* < 0.001, respectively.

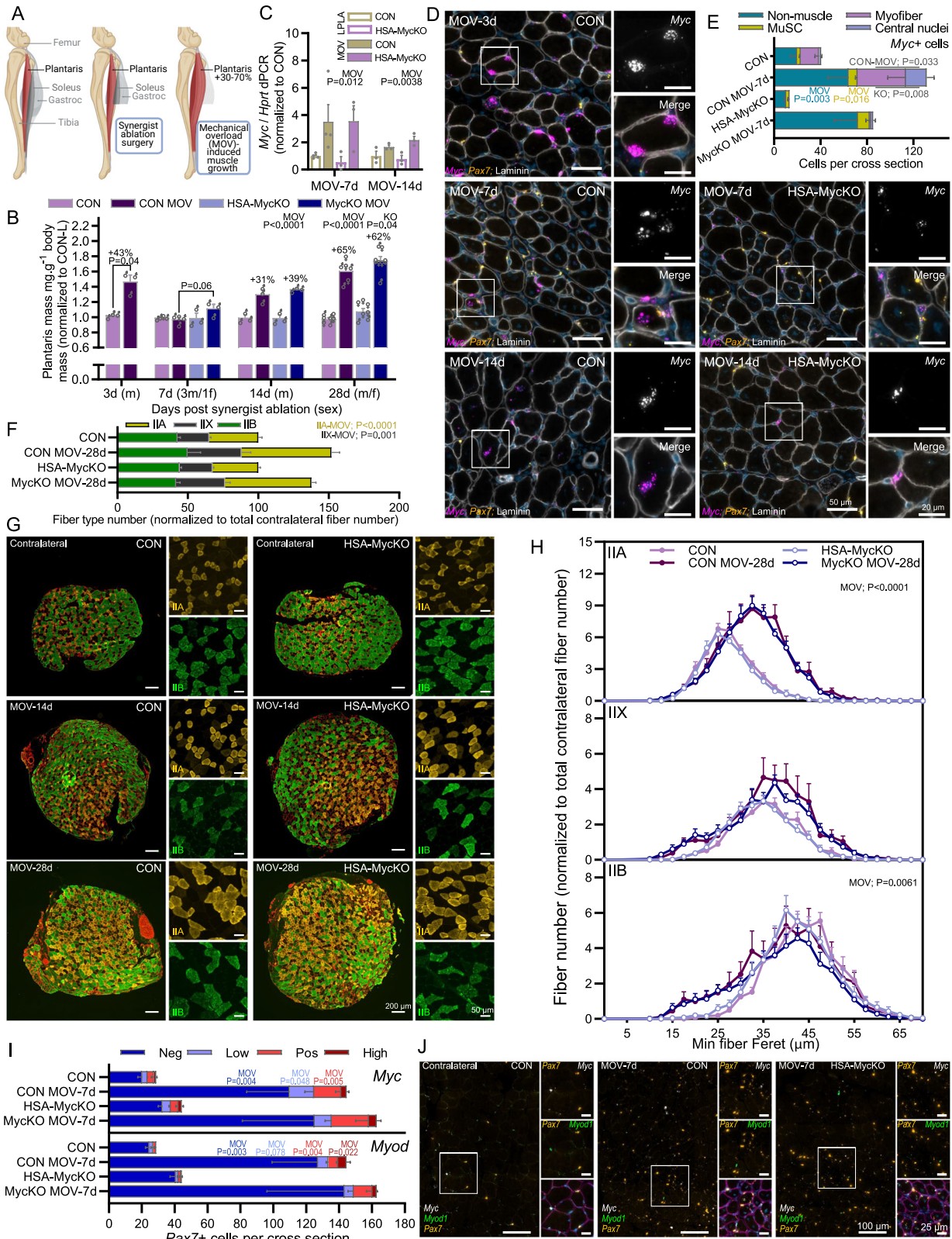

induced muscle growth, HSA-MycKO mice showed strong MOV-induced muscle growth equivalent to CON levels.

In whole-muscle lysates, *Myc* mRNA expression was strongly induced after 7 and 14 days of MOV and this response was unaltered in HSA-MycKO mice (Fig. 2C). To confirm that, in our hands, MOV increases *Myc* mRNA expression in muscle fibers and that this is efficiently depleted in HSA-MycKO mice, we performed RNA-FISH with

probes against *Myc* and *Pax7* and counterstained with DAPI and laminin antibodies. Many muscle fiber nuclei expressing at least 3 *Myc* puncta (i.e. *Myc+*), as well as many non-muscle fiber cells, could be seen in PLA cross-sections from the ablated leg 3, 7 and 14 days after synergist ablation (Fig. 2D). While muscle fiber *Myc* expression was almost completely absent in HSA-MycKO mice at 7 and 14 days, many non-muscle fiber cells showed high *Myc* expression. Quantification of

**Fig. 2 | Mechanical overload- (MOV) induced muscle growth is not altered in HSA-MycKO mice. A** Schematic of MOV surgery. **B** *Plantaris* (PLA) muscle mass normalized to body mass and the mean mass of the contralateral muscle after 3 (CON only; $n = 3$), 7, 14 and 28 days of MOV in CON ($n = 4, 3, 7$) and HSA-MycKO ($n = 3, 3, 7$) mice. **C** Digital PCR for transcript levels of *Myc* and *Rpl3* normalized to a housekeeping gene (*Hprt*) in CON (HSA^wt/wt^; Myc^fl/fl^) and HSA-MycKO (HSA^Cre/wt^;Myc^fl/fl^) mice after 7 (CON, $n = 4$; KO, $n = 3$) and 14 ($n = 3$) days. **D** Representative images of RNA fluorescent in situ hybridization (RNA-FISH) with probes against *Myc* (magenta) and counterstaining with DAPI (blue) and laminin β1γ1 antibodies (white), for 3 (CON only), 7 and 14 days of MOV for CON and HSA-MycKO mice. **E** Quantification of *Myc* + (≥3 *Myc* puncta) nuclei outside laminin-stained muscle fibers (Non-muscle), *Pax7*+ muscle stem cell nuclei (MuSC) and peripherally (Myofiber) and centrally located (Central nuclei) nuclei within laminin stained muscle fibers, in contralateral and after 7 days of MOV (MOV-7d) in CON ($n = 4$) and

HSA-MycKO ($n = 3$) mice. **F** Type-specific (IIA, IIX, IIB) fiber number in complete PLA cross-sections normalized to the contralateral PLA in CON ($n = 6$) and HSA-MycKO ($n = 7$) mice after 28 days of MOV and (**G**) representative images after 14 and 28 days of MOV. **H** Type-specific (IIA, IIX, IIB) fiber size distribution normalized to contralateral total fiber number ($n = 7$; or 6 for CON MOV-28d). **I** Quantification and (**J**) representative images of *Pax7*+ nuclei co-stained with <3 (low) 3–5 (Pos) or >5 (High) RNA puncta for *Myc* (upper) and *MyoD* (lower) using RNA-FISH in the contralateral muscle and after 7 days MOV (CON, $n = 4$; KO, $n = 3$). **A** Created in BioRender. Ruegg, M. (2025) https://BioRender.com/d74q573. Data are presented as mean ± SEM. Two-way repeated measure ANOVAs (**A**) or two-way ANOVAs with Sidak's post hoc test were used to compare between data. *, **, and *** denote a significant difference between groups of $P < 0.05$, $P < 0.01$, and $P < 0.001$, respectively.

*Myc*+ cells in ablated and contralateral PLA muscle 7 days after synergist ablation in CON and HSA-MycKO mice showed substantial increases in the number of *Myc*+ nuclei in the periphery (Myofiber) and the center (Central nuclei) of the muscle fiber, as well as *Pax7*+ MuSCs and non-muscle cells located between muscle fibers (Fig. 2E). While the number of *Myc*-expressing non-muscle fiber cells and *Pax7*+ MuSCs were not altered in HSA-MycKO mice, the number of combined peripheral and centralized *Myc*+ nuclei in muscle fibers was ~96% lower, again demonstrating the efficient, muscle fiber-specific depletion of *Myc* in HSA-MycKO mice.

As muscle mass can also be influenced by edema, we measured type-specific cross-sectional fiber size 14 (Fig. S2B) and 28 (Fig. 2F–H) days after synergist ablation. MOV increased the absolute number of IIA and unstained (IIX) muscle fibers within complete, mid-belly cross-sections from the PLA muscle (Fig. 2F, G) after 28 days MOV, with no genotype effect. Mean minimum fiber Feret diameter increased in IIA muscle fibers, while IIB fibers shrank slightly after 14 (Fig. S2B) and 28 days (Fig. 2H) MOV, with no genotype effect. While increased fiber size did contribute to MOV-induced muscle growth, the contribution was mild compared to the increased fiber numbers present in cross-sections, which could technically be due to either muscle fiber hyperplasia or increased muscle fiber length, such that more fibers cross the muscle mid belly[30,31]. In either case, MOV-induced increases in cross sectional fiber number is likely to rely on MuSC fusion to support muscle growth, in line with a strong increase in the number of centrally nucleated fibers after 14 and 28 days MOV (Fig. S2A). Furthermore, using RNA-FISH with probes against *Pax7*, *Myod1* and *Myc*, we observed an ~4-fold increase in the number of *Pax7*+ cells, and an equivalent increase in *Pax7*+ cells expressing 3–5 (*Myc* Pos) and >5 (*Myc* High) *Myc* puncta in the ablated compared to the contralateral leg, 7 days after synergist ablation in both CON and HSA-MycKO mice (Fig. 2I, J). Similarly, the number of *Pax7*+ cells co-expressing 3–5 (*MyoD* Pos) and >5 (*MyoD* High) *MyoD* puncta were significantly increased 7 days after ablation in both CON and HSA-MycKO mice, indicating pronounced MuSC activation, proliferation and differentiation in response to MOV (Fig. 2I, J). Together, these data indicate that muscle fiber Myc expression is not necessary for MOV-induced muscle growth. However, MuSCs appear to be strongly involved in MOV-induced muscle growth and hence may compensate for the lack of muscle fiber Myc expression.

### Muscle stem cell Myc expression is necessary for mechanical overload-induced muscle growth

Given the 4-fold increase in MuSC content (Fig. 2I) and the 40–50% increase in cross-sectional muscle fiber number (Fig. 2F) in response to MOV, we wondered whether Myc expression in MuSCs would be necessary for MOV-induced muscle growth. To this end, we first bred a tamoxifen inducible Cre^ERT2^ (Pax7-Cre^ERT2^) mouse[32] and an eGFP reporter mouse[33] with floxed Myc mice to create Pax7^Cre/wt^; eGFP^ki/wt^; Myc^fl/fl^ (Pax7-MycKO) and Pax7^Cre/wt^; eGFP^ki/wt^; Myc^wt/wt^ (Pax7-eGFP)

mice. As Myc is known to play a central role in cell proliferation, including in muscle progenitor cells[27], we compared proliferation competence in Pax7-eGFP and Pax7-MycKO MuSCs by isolating single fibers from mice that were injected for 5 consecutive days with tamoxifen (Fig. 3A). Single fibers were either fixed immediately (0 h) or cultured in growth medium for 48 h (48 h) and then stained for Pax7 and Ki67, a marker of cell cycle activation (Fig. 3A, B). Even though the isolation process can disturb MuSC quiescence, just one third of eGFP+ cells on fibers from Pax7-eGFP mice were Ki67+ at 0 h, while 48 h of culture in growth medium increased the percentage of Ki67 + , eGFP+ cells to 94% (Fig. 3C). Similarly, 92% of eGFP+ cells from Pax7-eGFP mice at 0 h existed as isolated cells on muscle fibers, while 48 h of culture significantly increased the percentage of eGFP+ cells in clusters of 2, 3-4 and 5+ cells at 48 h (Fig. 3B, D), indicating strong MuSC proliferation. In contrast, Myc depletion severely blunted MuSC proliferation, with just 38% of eGFP+ cells expressing Ki67 and >80% existing as isolated cells on muscle fibers from Pax7-MycKO mice cultured for 48 h in growth media.

Next, to test the efficiency of MuSC Myc depletion in vivo, we initiated a degeneration-regeneration event in Pax7^wt/wt^; Myc^fl/fl^ (CON) and Pax7^Cre/wt^; Myc^fl/fl^ (Pax7-MycKO) mice via intramuscular injection of cardiotoxin (CTX), a model in which (1) regeneration depends on muscle-resident MuSCs and (2) Myc depletion in MuSCs has been shown to strongly impair regeneration[26]. To induce Myc depletion in MuSCs, CON and Pax7-MycKO mice were tamoxifen treated for 7 days prior to cardiotoxin injection and then 3 times per week thereafter to avoid any potential expansion of a non-recombined MuSC pool (Fig. 3E). TA muscle mass was significantly reduced 4 and 7 days post CTX injection in both CON and Pax7-MycKO mice before recovering in CON, but not Pax7-MycKO mice 14 days after injury (Fig. 3F). Four days after injury, the TA muscle of CON mice showed many small embryonic myosin heavy chain (eMHC) positive fibers, which grew and matured to primarily express mature (IIA/X and IIB) MHC isoforms 14 days after injury (Fig. 3G, H). While ~5% of fibers expressed eMHC and ~50% expressed IIB MHC in CON muscles 14 days after injury, ~25% of fibers still expressed eMHC and just ~15% expressed mature IIB MHC in Pax7-MycKO mice (Fig. 3G), which were also small in size (Fig. 3I). In line with our observations in cultured single fibers and the known role of Myc in cell proliferation, including muscle progenitor cells[27], the delayed regeneration observed in Pax7-MycKO mice was related to strong impairments in the expansion of the MuSC pool (i.e. Pax7+ cells; Fig. 3J) and subsequent differentiation (i.e. MyoG+ cells; Fig. 3K), thereby delaying fiber formation and leading to incomplete regeneration and continued presence of non-muscle cells at 14 days post injury (Fig. S3). These experiments show that tamoxifen injection into Pax7-MycKO mice results in highly efficient *Myc* deletion and confirm the importance of Myc for MuSC activation and proliferation as well as MuSC-mediated muscle regeneration.

To test whether Myc expression in MuSCs is necessary to support MOV-induced muscle growth, we treated CON and Pax7-MycKO mice

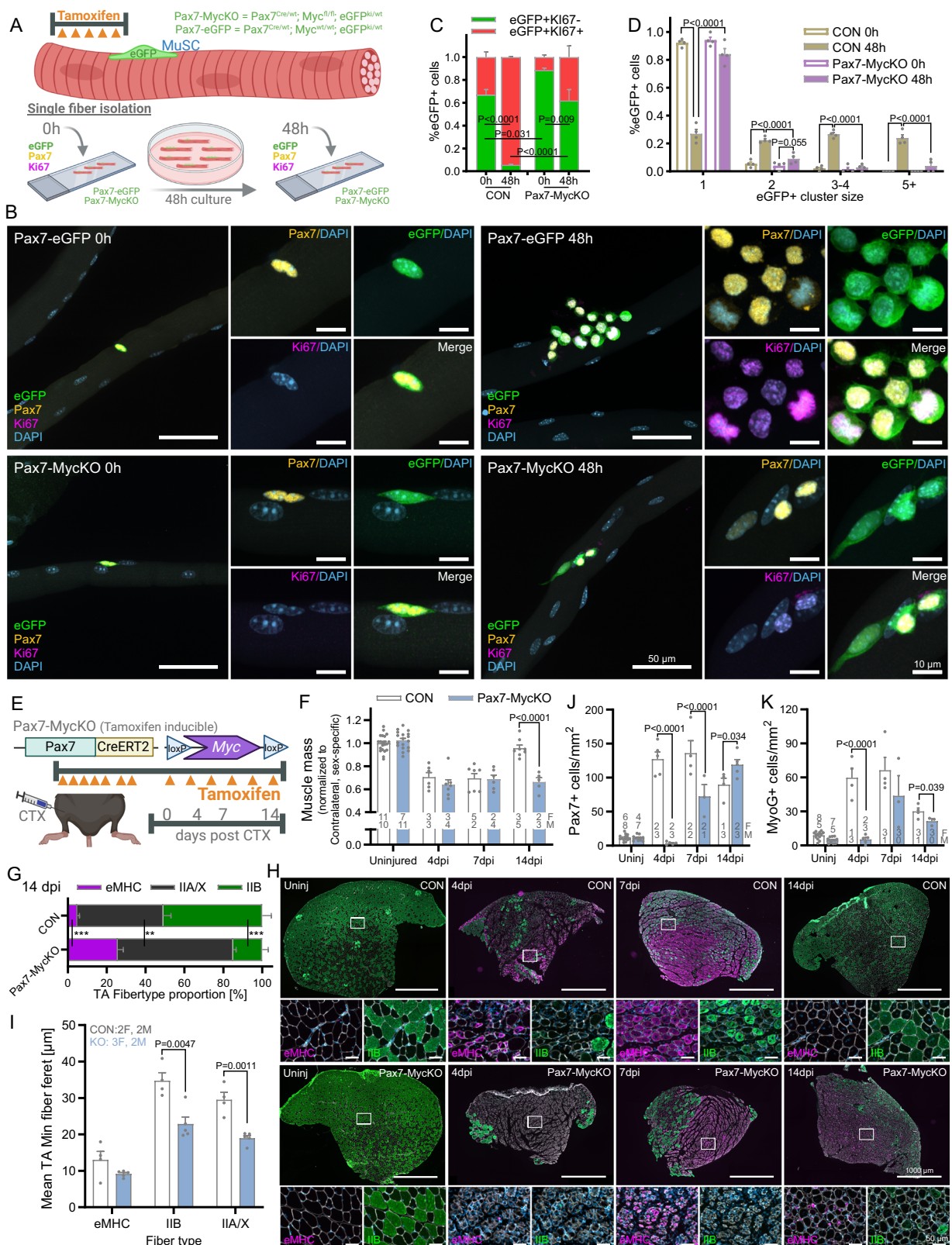

with tamoxifen according to the schedule described above (Fig. 3E) and performed synergist ablation surgery (Fig. 2A). PLA muscle mass increased 38 and 31% in female and male CON mice, respectively, after 28 days MOV compared to the contralateral leg (Fig. 4A). In contrast, PLA mass increased in female Pax7-MycKO mice by a significantly lower 11% after 28 days, while mass actually decreased by 10% in male Pax7-MycKO mice. While MOV-induced muscle growth was severely

curtailed in both sexes, due to the stronger impairment of muscle growth in male mice, we analyzed type-specific fiber size in female and male mice separately. Cross-sections from the mid-belly of PLA muscle confirmed the strong increase in whole-muscle cross-sectional area in CON mice, which was strongly impaired in Pax7-MycKO mice (Fig. 4B). In females, CON and Pax7-MycKO mice showed similar changes in type-specific fiber size distribution (Fig. 4C, upper), but female Pax7-

**Fig. 3 | Proliferation is blunted in Myc-depleted MuSCs and muscle regeneration is strongly impaired in Pax7-MyckO mice. A** Schematic and (**B**) representative images of isolated single fibers from Pax7-eGFP and Pax7-MycKO mice either fixed immediately (0 h) or cultured for 48 h (48 h) and then stained for eGFP, Pax7 and Ki67 (*n* = 4). **C** Percentage of eGFP+ MuSC cells expressing Ki67. **D** Percentage of eGFP+ MuSC cells in cluster sizes of 1, 2, 3-4 and 5+. **E** Schematic of cardiotoxin (CTX) experiment in CON and Pax7-MycKO mice. **F** TA Muscle mass 4, 7 and 14 days after CTX injury in CON (*n* = 6, 7, 8) and Pax7-MycKO mice (*n* = 7, 6, 5) normalized to uninjured contralateral TA mass. **G** Fiber type proportion for eMHC, IIB and unstained fibers (IIA/X) 14 days after CTX injury in CON (*n* = 4) and Pax7-MyckO (*n* = 5) TA cross-sections. **H** Representative images showing immunostaining for embryonic myosin heavy chain (eMHC; magenta), type IIB (green) and

laminin (white) at 4, 7 and 14 days post CTX injury and (**I**) mean minimum fiber feret size at 14 days in CON (*n* = 4) and Pax7-MycKO (*n* = 5) TA cross-sections. **J** Pax7+ and (**K**) MyoG+ cell number in TA muscle sections, via immunostaining, 4 (*n* = 4-5), 7 (*n* = 3-4) and 14 (*n* = 3-5) days after CTX injury in CON and Pax7-MycKO mice normalized to cross-sectional area. Uninjured samples represent pooled, un-injected contralateral legs from all time points (CON, *n* = 13–14; KO, *n* = 11–12). **A, E** Created in BioRender. Ruegg, M. (2025) https://BioRender.com/d74q573. Data are presented as mean ± SEM. Two-tailed, Student's independent *t* tests (**I**) or two-way ANOVAs with Sidak's post hoc test were used to compare between conditions. *, **, and *** denote a significant difference between groups of *P* < 0.05, *P* < 0.01, and *P* < 0.001, respectively.

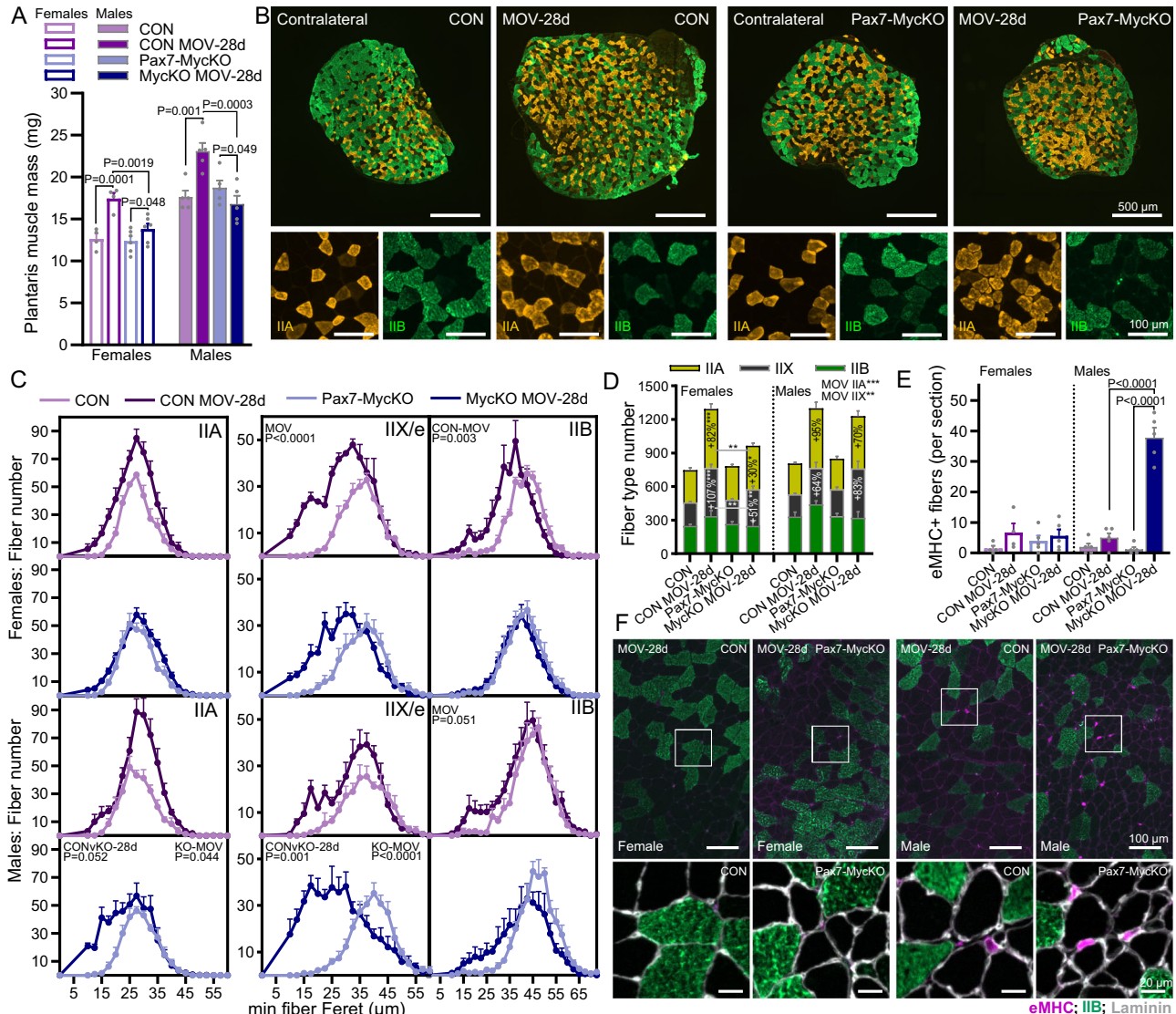

**Fig. 4 | MOV-induced muscle growth is strongly impaired in Pax7-MyckO mice. A** Muscle mass (CON, *n* = 4 F and 5 M; KO, *n* = 5 F and 5 M) and (**B**) representative images (male), (**C**) fiber size distribution and (**D**) total fiber number of *plantaris* (PLA) muscle cross-sections from the contralateral control limb and after 28 days of MOV in female (CON, *n* = 4; KO, *n* = 5 or *n* = 6 for KO-MOV) and male CON (*n* = 4) and Pax7-MycKO (*n* = 5) mice, stained with antibodies against IIA and IIB MHC isoforms. **E** Quantification of eMHC+ fibers in female and male CON (*n* = 4 F, 5 M)

and Pax7-MycKO (*n* = 5) PLA muscle cross-sections from the contralateral limb or after 28 days of MOV and (**F**) representative images of eMHC, IIB and laminin stained sections from female (left) and male (right) muscles 28 days after MOV. Data are presented as mean ± SEM. Mixed-effects (**A**) or two-way ANOVAs with Sidak's post hoc test were used to compare between conditions. *, **, and *** denote a significant difference between groups of *P* < 0.05, *P* < 0.01, and *P* < 0.001, respectively.

MycKO mice displayed significantly lower MOV-induced increases in cross-sectional fiber number (Fig. 4D, left). In contrast, male Pax7-MycKO mice displayed significantly smaller median IIA and IIX fiber size (Fig. 4C, lower), but increases in IIA and IIX fiber number comparable to CON mice (Fig. 4D, right). We also saw a conspicuous increase in the number of very small fibers unstained for IIA and IIB MHC isoforms in male but not female Pax7-MycKO mice. These fibers were positive for eMHC, indicating nascent fiber formation, albeit delayed, in male, but not female, Pax7-MycKO mice (Fig. 4E, F). Together, these results indicate that Myc expression in MuSCs is essential for a robust MOV-induced muscle growth response, with Myc-depleted MuSCs either failing to increase cross-sectional fiber number (females) or failing to support growth and maturation of nascent fibers (males).

## Myc is necessary to support MuSC fusion during muscle regrowth after a peripheral nerve injury

In both CTX-induced muscle regeneration and synergist ablation experiments, the overall phenotype observed in Pax7-MycKO mice is likely based on impaired proliferation of Myc-depleted MuSCs. To test whether Myc expression in MuSCs would also impair muscle growth under conditions where the proliferative burden on MuSCs is lower, we crushed the femoral branch of the sciatic nerve, which causes complete denervation of the lower hindlimb, followed by re-innervation at around 14 days and subsequent muscle regrowth[34], approaching completion after ~6 weeks. Loss of innervation stimulates MuSCs to activate and proliferate[35], presumably to support muscle regrowth and NMJ remodeling in the event of reinnervation[36]. Furthermore, to determine whether Pax7-MycKO MuSCs fuse with muscle fibers after a denervation/re-innervation event, we compared Pax7-eGFP and Pax7-MycKO mice, where all muscle fibers that incorporate MuSCs will become eGFP positive (Fig. 5A). Mice were given daily tamoxifen injections for 7 days prior to unilateral sciatic nerve crush and three times per week thereafter to avoid any potential expansion of a non-recombined MuSC pool (Fig. 5A). In both innervated GAS muscle and 14 days after crush, Pax7-eGFP mice showed very few eGFP + muscle fibers (Fig. S3A), indicating that short-term denervation alone is not sufficient to stimulate MuSC fusion with myofibers. On the other hand, many eGFP+ fibers could be seen in all tested muscles from Pax7-eGFP mice 28 d after nerve crush (i.e., after muscle fiber regrowth) with the muscle-specific prevalence pattern of eGFP+ fibers such that GAS > TA > EDL & SOL (Fig. 5B). MuSC fusion continued to contribute to ongoing muscle regrowth beyond 28 d as the presence of eGFP+ fibers further increased between 28 and 42 days after nerve crush (Fig. S3A). Strikingly, eGFP+ fibers were largely not detected in any muscle from Pax7-MycKO mice (Fig. 5B). Quantification at 28 days post crush showed that <3 fibers per mm$^2$ were eGFP+ in Pax7-MycKO mice whereas CON mice reached 106 eGFP+ fibers per mm$^2$ (Fig. 5C). The presence of eGFP+ fibers was also dramatically reduced in GAS muscle from Pax7-MycKO mice 42 d after nerve crush (Fig. S4A). Although this cytosolic eGFP lineage tracing method does not allow absolute quantification of MuSC-myofiber fusion events, the almost complete abolition of eGFP+ fibers in Pax7-MycKO mice supports the notion that Myc expression in MuSCs supports MuSC contributions to physiological muscle growth. Interestingly, eGFP+ fiber prevalence in Pax7-eGFP mice correlated with the extent of muscle mass loss, with GAS, TA, EDL and SOL losing 42, 33, 14 and 17% muscle mass (Fig. 5D), respectively, suggesting increasing MuSC fusion events with increasing requirements for muscle regrowth. However, in Pax7-MycKO mice, muscle mass recovery was identical to CON mice for all tested muscles. Furthermore, Myc depletion in MuSCs did not affect muscle fiber size in TA muscle cross-sections 14 or 28 days after nerve crush (Fig. S4B) or specific force in the EDL muscle (Fig. 5E) 28 and 42 days after nerve crush. As *Myc* mRNA expression peaked 14 days after nerve crush (Fig. 5F), we examined the effect of Myc depletion on regrowth-

induced MuSC proliferation by injecting mice with EdU 24 h before collecting tissue 14 d after nerve crush, thereby labeling all cells that replicated their DNA within this period. Nerve crush significantly increased (+54%) the number of Pax7+ MuSCs in whole GAS muscle cross-sections compared to contralateral limbs in CON mice, but not Pax7-MycKO mice (Fig. 5G, H). Similarly, the number of Ki67+ (Fig. 5G, I) and EdU+ (Fig. 5J) MuSCs were strongly elevated 14 d after nerve crush in CON but not Pax7-MycKO mice.

Next we quantified the number of eGFP+ and eGFP- centrally nucleated fibers, which result from muscle fiber regenerative events involving progenitor cell fusion[37]. Pax7-eGFP and Pax7-MycKO mice displayed comparable numbers of centralized nuclei in TA cross-sections; however, while 65% of centralized nuclei co-localized with eGFP staining in Pax7-eGFP mice, just 2% were eGFP+ in Pax7-MycKO mice (Fig. 4I, J). These findings indicate that skeletal muscle can tolerate a lack of MuSC proliferation/fusion during muscle regrowth after nerve injury, perhaps by relying more heavily on fusion-competent, progenitor pools not expressing Pax7. Indeed, a population of Pax7-negative cells with myogenic potential has been identified within skeletal muscle, expressing markers of smooth muscle and mesenchymal cells (SMMCs)[38]. Furthermore, bone marrow-derived hematopoietic stem cells have also been shown to fuse to fibers and support the regeneration of injured muscle[39]. Therefore, contributions from such cells could explain the full recovery of muscle size in Pax7-MycKO mice after nerve crush.

## Muscle fiber Myc expression is not necessary for PKBα/Akt1-induced muscle hypertrophy or signaling

As MOV-induced muscle growth was heavily MuSC-dependent, we next investigated the possibility that muscle fiber Myc expression is necessary to support an increase in muscle fiber size, i.e. hypertrophy. Activation of the PKB/Akt-mTORC1 pathway in muscle fibers promotes robust hypertrophy[40]. In non-muscle cells, this pathway and Myc promote cell growth through intertwined and often co-dependent mechanisms, with Myc-driven cell growth depending on mTORC1 activity[41] and Akt-driven growth responsive to Myc activity[42]. Myc and the Akt-mTORC1 pathway have also been hypothesized to function cooperatively in skeletal muscle hypertrophy[43], but this has not been directly tested.

To test whether Myc is required for muscle fiber hypertrophy mediated by the Akt-mTORC1 pathway, we crossed AktTG mice expressing an active, myristoylated form of PKBα/Akt1, fused to eGFP and ERT2 in skeletal muscle[44] with HSA-MycKO mice to create AktTG and AktTG; HSA-MycKO (Akt-MycKO) mice (Fig. 6A). In these mice, the PKBα/Akt1-ERT fusion protein is constitutively expressed in skeletal muscle fibers but immediately degraded and hence not active. Injection of tamoxifen, which binds ERT, stabilizes the PKBα/Akt1-ERT fusion protein, rendering it constitutively active[45,46] and causing robust PKBα/Akt1-mediated muscle hypertrophy[44].

Just 3 days of tamoxifen injections was sufficient to promote measurable increases in QUAD, EDL, PLA, GAS and TRI muscle mass in female mice, while 14 days of tamoxifen treatment robustly increased the mass (25–75%) of all measured muscles, except the slow-twitch SOL muscle, which displayed just a minor ~6% increase in mass (Fig. 6B). Surprisingly, PKBα/Akt1-induced muscle hypertrophy was identical in AktTG and Akt-MycKO mice for both female (Fig. 6B) and male (Fig. 6C) mice, with the magnitude of muscle growth slightly higher in male than female mice. Importantly, AktTG-induced muscle growth was functional, with a significant upward shift in the force frequency curve for EDL in vivo muscle function measurements (Fig. 6D). Apart from a small MycKO main effect for reduced muscle force at 50 Hz stimulation frequency, the increase in EDL muscle function following 14 days of tamoxifen treatment was comparable between AktTG and Akt-MycKO mice (Fig. 6D). To confirm the typical IIB-specific fiber hypertrophy resulting from PKBα/Akt1 activation in AktTG and Akt-

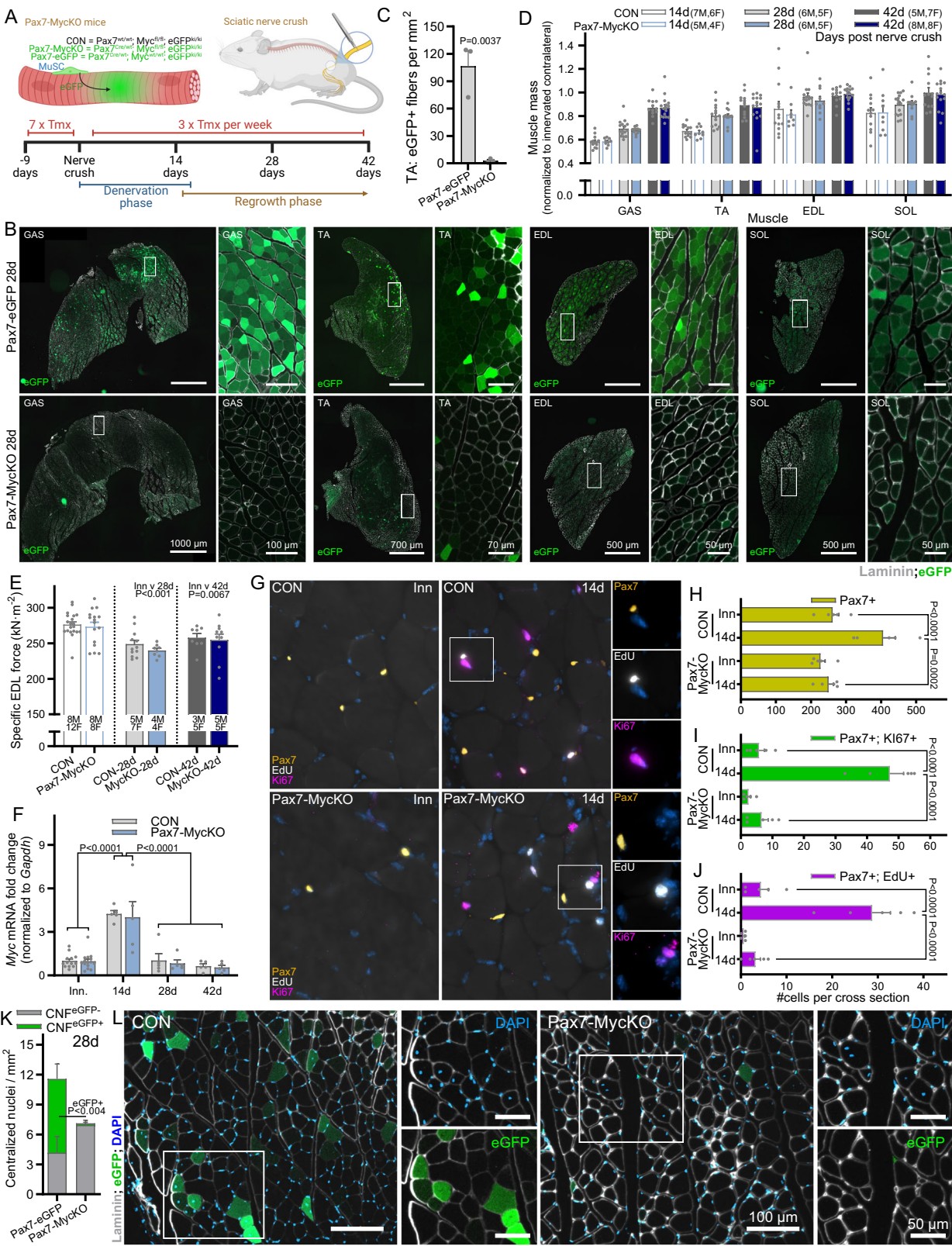

MycKO mice, we quantified type-specific fiber size in complete TA cross-sections (Fig. 6E). While fiber size distribution was unaffected by 14 days of PKBα/Akt1 activation in IIA and IIX fibers, IIB fiber size distribution showed an identical and marked shift to the right (i.e. larger fibers) in both AktTG and Akt-MycKO mice (Fig. 6F).

While muscle fiber Myc depletion is not sufficient to perturb PKBα/Akt1-induced hypertrophy, we wondered whether loss of Myc

would lead to more subtle alterations in the signaling pathways associated with rapid muscle hypertrophy. To this end, we performed mRNAseq on GAS muscles from female CON and HSA-MycKO mice as well as AktTG and Akt-MycKO mice after 3 and 14 days of tamoxifen treatment. PCA analysis showed clear separation into three sample groups on PC1 and PC2, with mice not expressing PKBα/Akt1 and those stimulated for 3 d and 14 d clustering together, independent of Myc

**Fig. 5 | Myc depleted MuSCs fail to contribute to muscle regrowth after sciatic nerve injury. A** Lineage tracing approach to track MuSC-myofiber fusion events following sciatic nerve crush. **B** Representative images of eGFP+ fibers, indicating Pax7+ MuSC fusion, in GAS, TA, EDL and SOL muscles of Pax7-eGFP mice, and their absence in Pax7-MycKO mice 28 days after nerve crush and (**C**) Quantification in TA muscle (n = 3). **D** Muscle mass 14, 28 and 42 days after sciatic nerve crush normalized to the innervated contralateral limb in CON (n = 13 or 12 for SOL, 16 or 15 for TA and EDL, and 13 or 12 for GAS and EDL) and Pax7-MycKO (n = 9 or 8 for EDL, 11 or 10 for EDL, and 15 or 16 for GAS) mice. **E** EDL specific muscle force 28 and 42 days after nerve crush and in the contralateral control muscle for CON (28 d, n = 12; 42 d, n = 8) and Pax7-MycKO (28 d, n = 8 or 6 Inn; 42 d, n = 10) mice. **F** RT-qPCR for *Myc*

mRNA normalized to *Gapdh* in innervated (CON, n = 13; KO, n = 15) muscle and 14, 28 and 42 days after nerve crush (n = 5). **G** Representative images and quantification of (**H**) Pax7+ (**I**) Pax7+; Ki67+ and (**J**) Pax7+ ; EdU+ cell number in immunostained whole GAS muscle cross-sections collected 14 d after nerve crush and 24 h after an I.P. injection with 100 mg/kg EdU (n = 5). **K** Quantification and (**L**) representative images of eGFP+ and eGFP- fibers with centralized nuclei in TA muscle from Pax7-eGFP and Pax7-MycKO mice (n = 3). **A** Created in BioRender. Ruegg, M. (2025) https://BioRender.com/d74q573. Data are presented as mean ± SEM. Two-tailed Student's *t* tests (**C**) or two-way ANOVAs with Sidak's post hoc test (**D–F, H–K**) were used to compare between conditions. *, **, and *** denote a significant difference between groups of $P < 0.05$, $P < 0.01$, and $P < 0.001$, respectively.

expression (Fig. 6G). Compared to CON mice, PKBα/Akt1 activation differentially regulated (FDR < 0.05; Log fold change >0.5) 2418 genes after 3 d and 4221 genes after 14 d, with 3369 genes differentially regulated between 3 d and 14 d (Fig. 6H). In Akt-MycKO mice, PKBα/Akt1 activation differentially regulated a comparable number of genes after 3 d (2224) and 14 d (4405) compared to MycKO mice, while 3017 genes were differentially regulated between 3 and 14 d. While PKBα/Akt1 activation induced substantial changes in gene expression, changes between CON and HSA-MycKO mice were minor, with 56, 145 and 235 differentially regulated genes in mice not expressing the AktTG and those stimulated for 3 d and 14 d, respectively. In addition to the comparable magnitude of PKBα/Akt1-induced gene expression changes between CON and HSA-MycKO mice, gene ontology terms (Biological process, Molecular function and Cellular component) were largely enriched (Fig. 6I) and depleted (Fig. 6J) to similar extents in CON and HSA-MycKO mice after 14 d PKBα/Akt1 activation, with the exception of terms related to ribosome biogenesis, which were more strongly depleted in AktTG than Akt-MycKO mice. Further examination of ribosomal protein gene expression showed strong downregulation in both 14 d AktTG and Akt-MycKO mice (Fig. S5A, B), compared to CON and HSA-MycKO mice, respectively. The fold change in ribosomal protein gene expression was also strongly correlated between these two comparisons, but the slope of the relationship indicated greater downregulation of gene expression in 14 d AktTG than Akt-MycKO mice (Fig. S5C). Together, these results indicate that Myc depletion in muscle fibers does not impair PKBα/Akt1-induced muscle hypertrophy or gene expression changes.

**Muscle fiber Myc overexpression perturbs muscle regeneration, re-growth and homeostasis.** While our experiments conclusively demonstrate that muscle fiber Myc expression is not necessary for muscle homeostasis, post-natal growth or muscle fiber hypertrophy, it remains possible that acute muscle fiber expression of Myc is sufficient to promote muscle growth. To explore this possibility, we crossed mice that express human Myc under the CAG promoter after removal of a floxed STOP cassette[47]. When mated with mice expressing either Pax7-Cre[ERT2 32] or HSA[MerCreMer 48], mice will overexpress Myc after tamoxifen treatment in MuSCs (Pax7-MycTG) or muscle fibers (HSA-MycTG), respectively (Fig. 7A, E).

As muscle regeneration was strongly impaired in Pax7-MycKO mice, we first tested whether overexpressing Myc in MuSCs would improve muscle regeneration following an intramuscular cardiotoxin injection (Fig. 7A). However, while CON mice displayed a typical pattern of muscle mass loss 4 (−17%) and 7 (−24%) days after injury, followed by restoration after 14 days, muscle mass loss was strongly exacerbated at 7 days (−45%) and reduced further (−55%) 14 days after injury in Pax7-MycTG mice (Fig. 7B). The severely impaired regenerative capacity in Pax7-MycTG mice did not appear to result from a failed expansion of the MuSC pool (Fig. S5D) or MuSC differentiation, since many eMHC+ fibers could already be seen 4 days after injury (Fig. 7C). While nascent fibers successfully form in regenerating muscle from Pax7-MycTG mice, they fail to grow and mature, with the majority (64%) still expressing eMHC and <5% expressing the mature IIB MHC

isoform 14 days after injury, compared to <5% eMHC and 51% IIB in CON mice (Fig. 7D). Together, these results indicate that rather than supporting muscle growth, continued Myc expression in newly formed muscle fibers is detrimental to their growth and maturation.

As the impaired growth of nascent muscle fibers in Pax7-MycTG mice may be related to impaired maturation, we next wanted to test the effect of Myc expression in mature muscle fibers. To this end, we administered 5 daily injections of tamoxifen to CON and HSA-MycTG mice and examined their muscles 4 or 10 days after the first injection (Fig. 7E). Myc protein abundance was strongly elevated in whole-muscle lysates from 10 d HSA-MycTG compared to CON mice, while some (e.g. RPS14) but not all (e.g. RPS6) ribosomal proteins were mildly elevated (Fig. 7F, G). Body mass was unaffected in 4 d HSA-MycTG mice, but significantly reduced in 10 d HSA-MycTG mice over the treatment period (Fig. 7H). Remarkably, just 4 days after the first tamoxifen injection, specific EDL muscle force was already strongly reduced (−20%) at a stimulation frequency of 200 Hz and continued to rapidly deteriorate, reaching -50% at 10 days (Fig. 7I). In line with the rapid functional deterioration, strong pathological changes were visible in TA muscle cross-sections from 10 d HSA-MycTG mice, including atrophic and necrotic fibers as well as marked mononuclear cell infiltration (Fig. 7J, left). To confirm the loss of muscle fiber integrity seen in H&E stained HSA-MycTG muscle, we measured membrane permeability by staining for mouse IgG[49–51]. While IgG+ fibers could not be detected in CON muscle, 10 d HSA-MycTG mice displayed a marked elevation of IgG+ fibers (Fig. 7J, right), confirming loss of muscle fiber integrity. Not only did muscle fiber Myc expression destabilize muscle fibers, but it failed to increase muscle mass in all measured fast-twitch muscles (Fig. 7K). In contrast, the slow-twitch soleus muscle showed a small increase in mass in 10 d HSA-MycTG compared to CON mice. Together, these results indicate that Myc overexpression in muscle fibers, even for just a few days, is catastrophic for muscle fiber homeostasis.

As previous studies have shown Myc-induced pro-growth gene expression signatures, we wondered whether HSA-MycTG mice also shared gene expression patterns with PKBα/Akt1-mediated muscle hypertrophy, despite the stark difference in phenotypical outcomes. To this end, we performed mRNAseq on GAS muscle from 4 d and 10 d CON and HSA-MycTG mice. PC analysis showed clear separation of CON samples (4 d and 10 d) from HSA-MycTG samples (Fig. 8A). As 4 d and 10 d CON samples were indistinguishable from each other, we pooled these samples for further analysis. Robust gene expression changes were seen in both 4 d (4412 DEGs) and 10 d (6384 DEGs) HSA-MycTG mice, with the majority of DEGs (3273; 74%) at 4 d also differentially expressed in the same direction at 10 d (Fig. 8B). Top-enriched gene ontology terms (Biological process) in 4 d and 10 d HSA-MycTG muscle contained multiple terms associated with ribosome biogenesis, with the majority of ribosomal protein genes significantly increased (Fig. 8C), as well as "mitochondrial translation" and "methylation" (Fig. S5E). Similarly, genes associated with the terms "cell adhesion", "cell migration" and "ECM organization" were commonly downregulated in 4 d and 10 d HSA-MycTG muscle, while genes associated with "cell-matrix adhesion", "integrin-mediated signaling pathways"

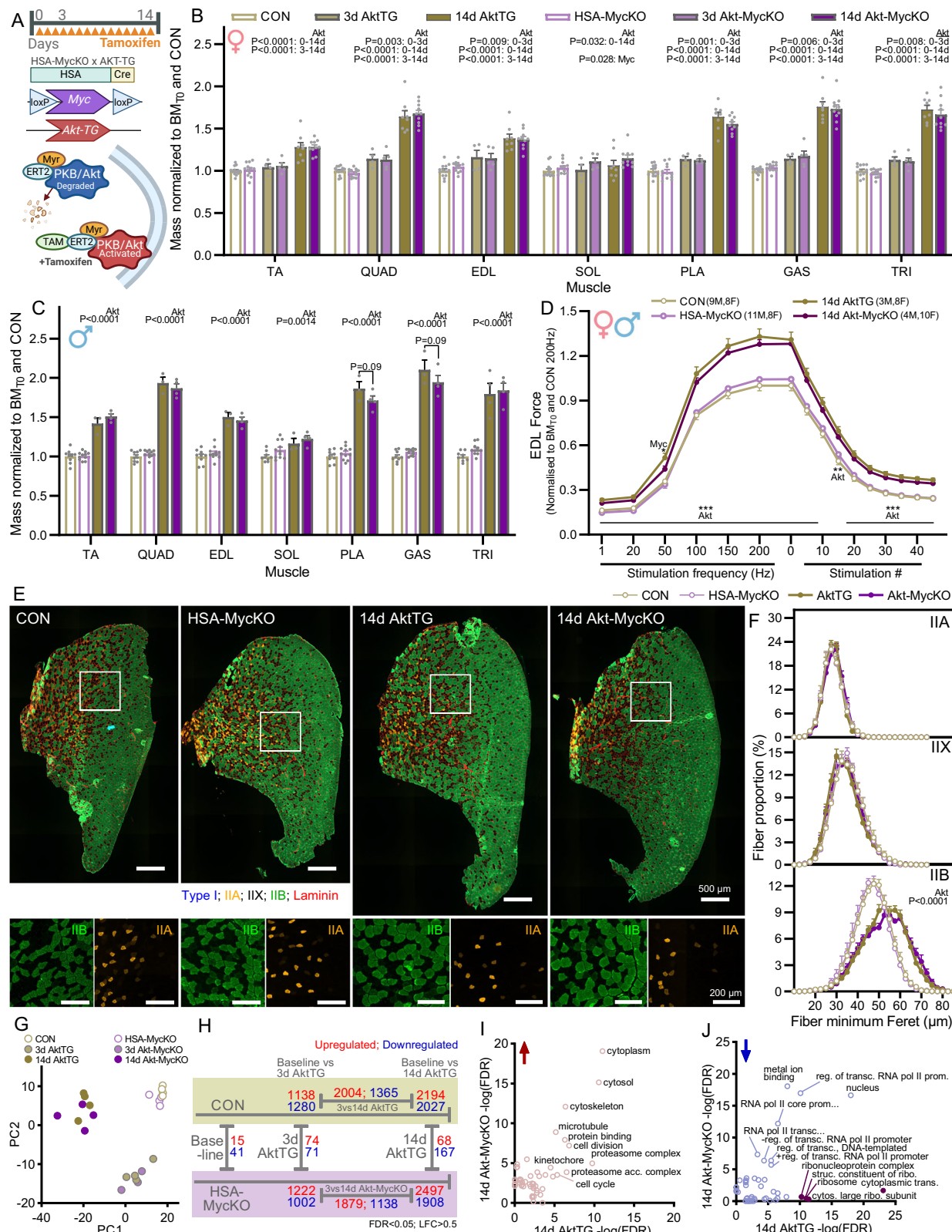

and "regeneration of osteoblast differentiation" (magenta) were particularly enriched at 4 days (Fig. S5F). At 10 days, "regulation of transcription from RNA polymerase II promoter" and "multicellular organism development" (all orange) were particularly downregulated (Fig. S5F).

In line with previous studies comparing MOV-induced gene expression signatures with muscle fiber Myc overexpression, we observed a strong overlap of differentially expressed genes under both acute (3 d AktTG and 4 d HSA-MycTG) and chronic (14 d AktTG and 10 d HSA-MycTG) activation (Fig. 8D). Genes commonly regulated in 10 d HSA-MycTG and 14 d AktTG mice were linked to regulation of transcription (downregulated) as well as cell proliferation and mitochondrial translation (upregulated; Fig. 8E). The major GO terms (biological process) oppositely regulated related to ribosome

**Fig. 6 | Muscle fiber Myc expression is not necessary for AktTG-induced muscle hypertrophy. A** Schematic of tamoxifen-inducible AktTG mice crossed with HSA-MycKO mice. Muscle mass for (**B**) female mice after 3 (CON, $n = 4$; KO, $n = 5$) and 14 (CON, $n = 8$; KO, $n = 10$) days and (**C**) male mice after 14 days (CON, $n = 3$ or 4 for TRI; KO, $n = 4$) of tamoxifen treatment. CON ($n = 13$ F, 9 M) and HSA-MycKO ($n = 12$ F, 11 M) mice treated with tamoxifen for 3 and 14 days but not expressing the PKBα/Akt1 transgene were pooled. **D** EDL muscle force (males and females; normalized to body mass prior to tamoxifen and force at a stimulation frequency of 200 Hz) for CON ($n = 17$), HSA-MycKO ($n = 19$), AktTG ($n = 11$) and Akt-MycKO ($n = 14$) mice treated with tamoxifen for 14 days. **E** Representative images and (**F**) quantification of type-specific fiber size distribution in TA muscle from female CON ($n = 7$), MycKO ($n = 7$), AktTG ($n = 7$) and Akt-MycKO ($n = 6$) mice after 14 days of tamoxifen (IIA: orange; IIB green; IIX unstained). **G** Principle component analysis of mRNAseq data generated from GAS muscle of female CON ($n = 4$) and HSA-MycKO ($n = 4$) mice treated with tamoxifen for 3 days or AktTG and Akt-MycKO mice treated for 3

(AktTG, $n = 4$; Akt-MycKO, $n = 3$) and 14 days ($n = 4$). **H** Differentially expressed genes (FDR < 0.05, Benjamini-Hochberg, or $P = 0.02$ for 3 d comparison due to low sample number; LFC > 0.5; up in red, down in blue) between CON and HSA-MycKO (baseline, i.e. no PKBα/Akt1) and either 3 or 14 days PKBα/Akt1 activation and between 3 and 14 days PKBα/Akt1 activation for CON and HSA-MycKO mice, as well as between CON and HSA-MycKO mice at baseline, 3 and 14 days. Scatterplot of -Log10(FDR) for (**I**) upregulated and (**J**) downregulated gene ontology (GO) terms (Biological process, Molecular function and Cellular component) enriched in AktTG (x-axis) or Akt-MycKO (y-axis) mice after 14 days tamoxifen compared to CON and HSA-MycKO mice, respectively. Top GO terms are labeled, while terms more prominently represented in AktTG than Akt-MycKO are shown in purple. **A** Created in BioRender. Ruegg, M. (2025) https://BioRender.com/d74q573. Data are presented as mean ± SEM. Two-way ANOVAs with Sidak's post hoc test were used to compare between conditions. *, **, and *** denote a significant difference between groups of $P < 0.05$, $P < 0.01$, and $P < 0.001$, respectively.

biogenesis, translation and p53 signaling (Fig. 8E), which were all downregulated in AktTG mice and upregulated in HSA-MycTG mice. Although the expression of ribosomal protein genes was suppressed in AktTG mice, translation of these genes is potently upregulated by PKB/Akt-mTORC1 pathway activation, resulting in higher ribosomal protein abundance[44]. Hence, despite the strong overlap of gene regulatory networks associated with cell growth, as previously observed[8], the resulting effect of Myc expression on post-mitotic muscle fibers is catastrophic, whereas PKBα/Akt1 activation results in functional hypertrophy. These juxtaposed phenotypic outcomes coincided with expression of muscle differentiation and maturation markers. For example, 3 d of PKBα/Akt1 activation promoted the expression of *Myod1* and *Mef2c*, key transcription factors promoting muscle differentiation[52], along with mature muscle markers such as *Myh4*, *Myl1* and *Acta1*. On the other hand, 4 d of MycTG expression strongly reduced the expression of the muscle differentiation transcription factor *Mef2c* as well as *Maf*[53], a transcription factor promoting maturation of fast-type skeletal muscle, along with many genes encoding for key muscle (particularly fast-type) proteins, including *Myh1*, *Myh4*, *Tnnc2*, *Tnni2*, *Mylpf*, *Myl1*, and *Tnnt3*.

Given that HSA-MycTG mice express extremely high levels of Myc, we next asked whether muscle fibers would tolerate and even grow when Myc is expressed at lower levels within the muscle. To examine this possibility, we took advantage of the mild MuSC fusion observed during the regrowth phase following a nerve crush injury to create a situation where just a few muscle fibers overexpress Myc (Fig. 8F). To track fusion of Pax7-MycTG MuSCs to muscle fibers, we crossed Pax7-MycTG mice with Pax7-eGFP (Fig. 5A) mice[33] to create Pax7$^{Cre/wt}$; eGFP$^{Kl/wt}$; MycTG$^{TG/wt}$ (Pax7-MycTG) mice and compared them with Pax7-eGFP (Pax7$^{Cre/wt}$-eGFP$^{Kl/Kl}$) mice (Fig. 8F). 28 days after nerve crush, TA muscle cross sections from Pax7-MycTG and Pax7-eGFP mice displayed a comparable elevation in the number of eGFP+ fibers (Fig. 8G, H). A small, non-significant shift in fiber size distribution was observed in eGFP+ compared to eGFP- fibers from Pax7-eGFP mice (Fig. 8I), while eGFP+ fiber size distribution displayed a strong shift towards smaller fibers compared to eGFP- fibers in Pax7-MycTG mice (Fig. 8J). Many of these small, eGFP+ fibers also displayed centrally located nuclei, which could be seen in both cross-sections and longitudinal sections (Fig. 8K) where small, localized chains of centralized nuclei could be visualized, a sign of focal regeneration via progenitor cell fusion[54,55].

Based on the low prevalence of eGFP+ fibers (~1–2%), muscle mass was unsurprisingly not different between genotypes (Fig. 8L). To evaluate muscle function, we next measured force-frequency curves in EDL muscle 28 days after nerve crush and compared this with the contralateral, un-injured EDL muscle. In CON mice, maximal force had largely recovered by 28 d, while a denervation-induced slowing in the contraction profile was still apparent, evidenced by increased fatigue resistance and higher force summation at 50 Hz stimulation frequency (Fig. 8M; left). In stark contrast, force recovery in EDL muscle of Pax7-

MycTG mice was strongly impaired compared to both the innervated contralateral muscle and CON 28 d muscle at stimulation frequencies between 100 and 250 Hz (Fig. 8M; right). Together, these data show that Myc expression is dispensable for muscle fiber growth and that overexpression of Myc, even in a small number of fibers, is detrimental to muscle homeostasis.

## Discussion

Best known for its central role in cancer cell malignancy, where it is elevated in up to 70% of all human cancers[56], Myc expression confers huge competitive advantages for proliferating cells[57]. Using Pax7-MycKO mice, we confirmed that Myc is vital for MuSCs to mount an appropriate myogenic response to regenerative or growth pressure[26], concomitant with strong impairments in MuSC activation and proliferation ex vivo (Fig. 3) and in vivo (Fig. 5). Even minor MuSC contributions to muscle regrowth were almost completely abrogated in Pax7-MycKO mice. On the other hand, transgenic Myc expression was not sufficient to expand the MuSC pool in injured or uninjured muscle (Fig. S5) or promote spontaneous fusion events (Fig. 7), indicating endogenous Myc expression is sufficient to support MuSC proliferation, differentiation and fusion in vivo.

Myc expression rapidly decreases in differentiated cells, but is re-expressed upon growth stimuli in many tissues, including skeletal[1] and cardiac muscle[58]. These observations, among others, led to the appealing hypothesis that muscle fiber Myc expression mediates a coordinated pro-growth response to anabolic stimuli, such as exercise[1,8,20]. However, unlike observations in cardiac muscle, where Myc is (1) required for stress-induced growth[58] and (2) transgenic Myc expression robustly promotes hypertrophy[59], our data paint a starkly different picture of Myc's role in skeletal muscle fiber growth and homeostasis. HSA-MycKO mice displayed normal post-natal muscle growth and responded effectively to both MOV- and AktTG-induced muscle growth stimuli, indicating muscle fiber Myc is dispensable for muscle growth and homeostasis. While MOV-induced muscle growth relied heavily on contributions from MuSCs, which did require Myc expression (Fig. 4), Akt-induced muscle growth is primarily based on an increase in the cytosolic volume of muscle fibers[60]. Furthermore, not only did transgenic skeletal muscle fiber Myc expression fail to promote hypertrophy, but it rapidly and profoundly impaired muscle fiber structure and function, even when expressed in a minority of muscle fibers (Fig. 8H–M). These data challenge the current paradigm and underscore a need to rethink the role of Myc in skeletal muscle fibers.

Using RNA-FISH, we confirmed previous assertions that MOV substantially increases muscle fiber Myc expression[8]. However, despite a > 95% reduction in *Myc*-expressing myonuclei, *Myc* mRNA was not lower in total muscle lysates from HSA-MycKO than CON mice after 7 d of MOV (Fig. 2), indicating muscle fibers are only minor contributors to the MOV-induced increase in muscle tissue *Myc* expression. On the

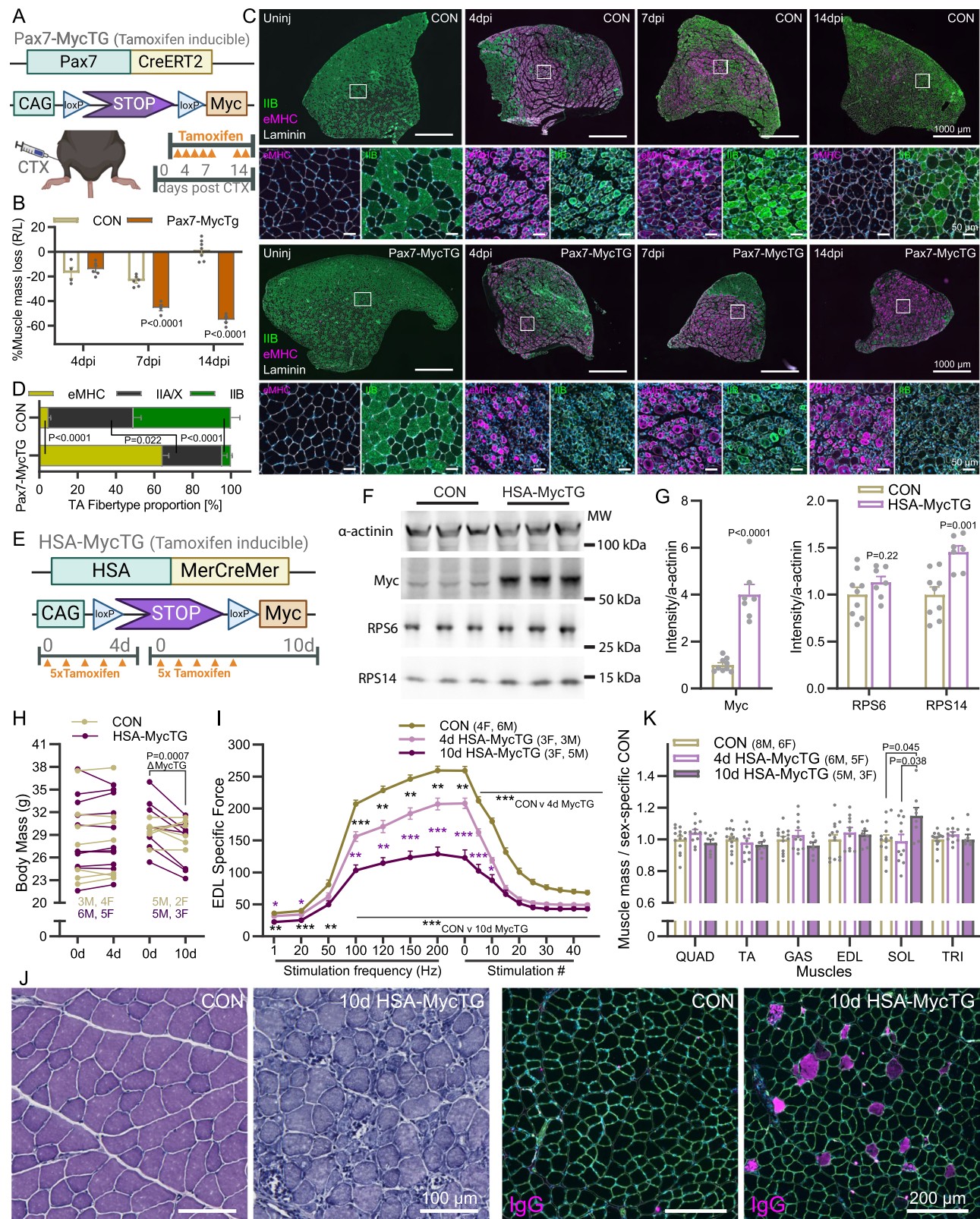

other hand, basal *Myc* expression in 2- and 7-month-old male HSA-MycKO mice was approximately halved (Fig. 1), suggesting muscle fibers are a significant source of whole-muscle *Myc* expression in sedentary mice. While these data support the reactivation of muscle fiber *Myc* expression in response to MOV, this response appears superfluous for muscle growth. A conceivable explanation for Myc's apparent dispensability in MOV-induced muscle growth could be that

fusing MuSCs provide the muscle fiber with the requisite components (e.g. RNA, proteins, organelles) needed to support muscle growth. The human skeletal α-actin (HSA) promoter is expressed in somites during embryogenesis (E 9.5) and in muscle progenitor cells upon commitment to the myogenic lineage[61], meaning recombination of floxed Myc alleles occurs prior to progenitor cell fusion. Although the short half-life of Myc protein means progenitor cells are unlikely to bring with

**Fig. 7 | Muscle fiber Myc expression strongly impairs muscle structure and function. A** Schematic of tamoxifen-inducible, muscle stem cell-specific transgenic Myc expression mouse model and time course of tissue collection after an intra-muscular cardiotoxin (CTX) injection. **B** Muscle mass loss (%) in CON and Pax7-MycTG mice 4 (CON, $n = 4$; TG, $n = 5$), 7 (CON, $n = 6$; TG, $n = 4$) and 14 (CON, $n = 7$; TG, $n = 5$) days after intramuscular CTX injection. **C** Representative images of eMHC, IIA/X (unstained) and IIB fiber type proportion in TA cross-sections 4, 7 and 14 days after intramuscular CTX injection and the contralateral control muscle (Uninj) and (**D**) quantification at 14 days ($n = 4$). **E** Schematic of HSA-MycTG mouse model and tamoxifen treatment regime. Representative western blots (**F**) and quantification (**G**) for Myc, RPS6 and RPS14, normalized to α-actinin (CON, $n = 9$; TG, $n = 7$). **H** Body mass immediately before the first tamoxifen injection and 4 or 10 days later for CON ($n = 7$) and HSA-MycTG (4 d, $n = 11$; 10 d, $n = 8$) mice. **I** Force frequency curve and fatigue response to repeated 200 Hz stimulations in EDL muscle from CON ($n = 10$) and HSA-MycTG mice at 4 ($n = 6$) or 10 d ($n = 8$) time points. **J** Representative images of H&E (left) and IgG (right) staining in TA cross-sections of CON and 10 d HSA-MycTG mice. **K** Muscle mass for CON ($n = 14$) and 4 d ($n = 11$) and 10 d ($n = 8$) HSA-MycTG mice. **A**, **E** Created in BioRender. Ruegg, M. (2025) https://BioRender.com/d74q573. Data are presented as mean ± SEM. Two-way ANOVAs (**B**, **D**, **I**, **K**) and repeated measures ANOVAs (**H**) with Sidak's post hoc test or Students $t$ test (**G**) were used to compare between conditions. *, **, and *** denote a significant difference between groups of $P < 0.05$, $P < 0.01$, and $P < 0.001$, respectively.

them meaningful Myc activity, it is possible that the transcriptional shadow of prior Myc activity lingers in muscle fibers. Indeed, many Myc-expressing myonuclei following MOV in CON mice were centralized (Fig. 2), perhaps indicating some level of residual Myc expression following progenitor cell fusion. However, signs of increased progenitor cell fusion (i.e. centralized nuclei) were not observed following PKBα/Akt1-mediated hypertrophy, suggesting muscle fibers are entirely capable of robust hypertrophy independent of Myc or its transcriptional shadow.

While the Akt-mTORC1 pathway and Myc activate overlapping cell growth pathways, they also cooperate and the activity of both have been shown to be obligatory for tissue/tumor growth mediated by proliferative cells. Indeed, Akt-mTORC1 signaling increases Myc protein abundance through enhanced translation efficiency[62] and protein stabilization[63] and inhibiting Myc strongly blunts tumor growth[42]. Similarly, Myc-driven tumor growth relies on mTORC1 signaling[41]. MuSC-mediated myogenesis is also severely impaired by loss of either Akt-mTORC1 signaling[64] or Myc (Figs. 3–5). The similar gene expression signatures observed in MycTG and AktTG mice (Fig. 8A–E) indicate that Akt-mTORC1 and Myc signaling also overlap in terminally differentiated skeletal muscle fibers. In contrast, we provide the first evidence that Myc is not required for robust PKB/Akt-mediated skeletal muscle fiber hypertrophy, suggesting alternate, Myc-independent pro-growth processes in skeletal muscle fibers. PKBα/Akt1 activation expands ribosomal protein content translationally via mTORC1[65,66], while ribosomal RNA synthesis can be mediated via casein kinase II alpha (*Csnk2a1*) and transcription initiation factor I (TIF1A), encoded by the *Trim24* gene[67], perhaps obviating the need for Myc activity in muscle fiber hypertrophy.

Although muscle fiber Myc expression appears dispensable for muscle fiber growth and homeostasis, the robust transcriptional response to transgenic Myc expression indicates that myonuclei are still supremely sensitive to changes in Myc levels. Almost 30% of the ~16k detected genes were significantly altered (LFC > 0.5; FDR < 0.05) in HSA-MycTG mice just 96 h after the first tamoxifen injection (Fig. 7). While Myc is thought to directly regulate 10–15% of all genes[68], at high expression levels, as seen in some tumors or upon experimental overexpression, Myc can also bind lower affinity gene targets and may regulate the majority of the genome[2]. Prolonged overexpression of Myc, which causes a rapid decline in homeostasis, is also likely to have contributed to the extent of gene expression changes in MycTG mice. Indeed, 10 days after the first tamoxifen injection, the proportion of genes with altered expression expanded to ~40% of detected genes in HSA-MycTG mice.

In contrast to the profound transcriptional changes in HSA-MycTG muscle, less than 1.5% of genes were significantly altered in muscles of 7-month-old male HSA-MycKO mice (Fig. 1) and even fewer (< 0.4%) in 4-month-old female HSA-MycKO mice (Fig. 5), indicating a limited effect of endogenous muscle fiber Myc on basal gene expression and muscle fiber homeostasis. A possible explanation for the dispensability of Myc in skeletal muscle fibers could be compensatory upregulation of the Myc family paralogs *Mycn* (N-Myc)

or *Mycl* (L-Myc), which share high sequence similarity with *Myc* (i.e. C-Myc) and have been shown to functionally compensate for loss of *Myc*, when appropriately expressed[69]. However, *Myc*, *Mycn* and *Mycl* are located on different chromosomes (1, 2 and 8, respectively), allowing divergent transcriptional regulation and therefore distinct tissue and temporal expression patterns[70]. Indeed, whole-body knockout of either *Myc* or *Mycn* are lethal between embryonic day 9 and 11[71,72]; however, when the *Myc* coding sequence is replaced by the corresponding *Mycn* coding sequence, mice are born, reach adulthood and are able to reproduce[69]. Interestingly, skeletal muscle development was strongly impaired in mice expressing *Mycn* instead of *Myc*, indicating a specific reliance on *Myc* for skeletal muscle development. In line with distinct regulation of Myc-family expression, *Mycn* and *Mycl* expression was ~10-fold lower than *Myc* expression in skeletal muscle lysates from CON mice with no sign of compensatory upregulation in HSA-MycKO mice.

The necessity of Myc activity within a given tissue depends on the expression levels of transcription factors within the Max-like (Mlx) network[2]. The Myc and Mlx networks, together referred to as the 'extended Myc network', share considerable target gene overlap[2]. Upon glucose or other nutrient binding, the Myc-like proteins ChREBP (*Mlxipl*) and MondoA (*Mlxip*) translocate to the nucleus where they can heterodimerize with the Max-like protein Mlx and bind carbohydrate response elements (ChoREs) comprising tandem E-boxes separated by 5 nucleotides in their target genes[73]. Using an albumin-driven Cre (Alb), the regenerative potential of Alb-MycKO hepatocytes is indistinguishable from CON hepatocytes[74], while the regenerative potential of Alb-MlxKO hepatocytes is strongly impaired and further worsened by double KO of Myc and Mlx[75]. Therefore, a combination of low basal Myc levels in muscle fibers and constitutive expression of the Myc-like Mlx network components MondoA and ChREBP[76] could explain the underwhelming molecular, physiological and phenotypic effects of Myc depletion in HSA-MycKO mice. Nevertheless, several key genes related to muscle fiber (e.g. *Dmd*) and NMJ (e.g. *MuSK*) function along with several TGFβ-pathway components (e.g. *Dkk3*, *Tgfbr1*) were all reduced in muscle from HSA-MycKO mice, indicating Myc may still play some homeostatic role within skeletal muscle fibers.

While prevailing Mlx pathway activity may explain Myc's dispensability in skeletal muscle fibers, it does not explain the failure of transgenic Myc expression to promote muscle growth despite robust, hypertrophy-like transcriptional changes (Fig. 7). Along with Oct3/4, Klf4 and Sox2, Myc is a Yamanaka factor (OKSM) and a single Myc pulse in skeletal muscle has been shown to induce some gene patterns in common with OKSM-induced partial molecular reprogramming towards more pluripotent states[20]. Myonuclei in differentiated skeletal muscle focus much of their transcriptional efforts on producing the mRNAs associated with the structural and contractile proteins that constitute the majority of a muscle fiber. Perhaps indicating a Myc-driven reprogramming towards a less differentiated state, genes encoding the mature muscle transcription factors *Mef2c* and *Maf* along with key muscle structural proteins, including *Myh1*, *Tnnc2*, *Tnni2*,

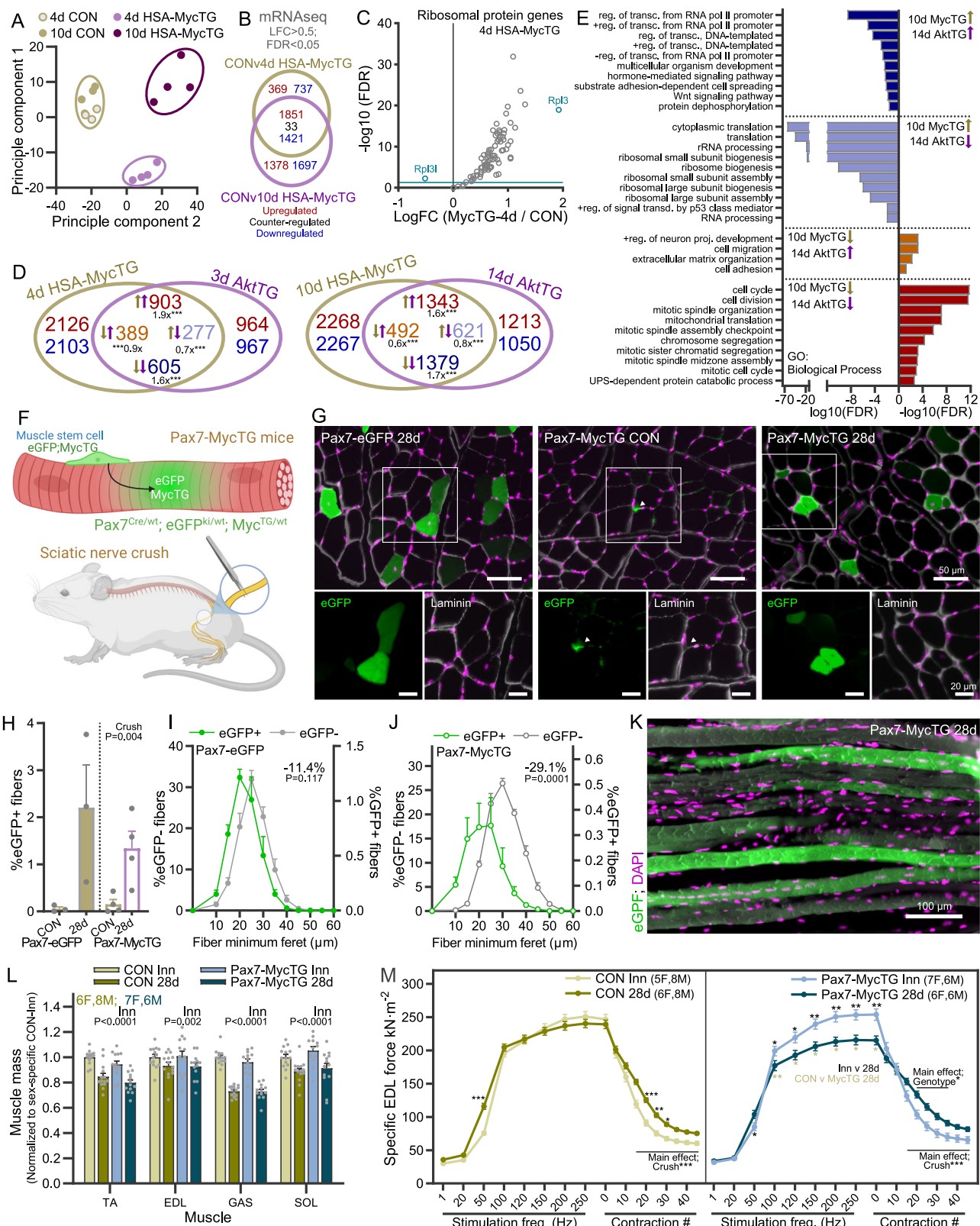

*Mylpf, Myl1, Tpm1, Tnnt3, Acta1* and *Myh4* were strongly suppressed in MycTG muscle just 96 h after the first tamoxifen injection. The widespread downregulation of mature muscle markers is suggestive of a 'dedifferentiation-like' response to Myc overexpression in muscle fibers. Myc-mediated dedifferentiation has previously been described in terminally differentiated epidermal cells during skin wound healing[77]. Not surprisingly, the outcome of shifting transcriptional focus away from characteristic mature muscle genes led to rapid disintegration of muscle fiber morphology and loss of muscle force.

Together, our data conclusively demonstrate that Myc is neither necessary nor sufficient to promote skeletal muscle hypertrophy. Rather than playing a central role in the muscles' response to anabolic stimuli, continued muscle fiber Myc expression is likely to impair skeletal muscle structure and function.

**Fig. 8 | Muscle fiber Myc overexpression promotes growth-related gene expression signatures, but even a minority of Myc overexpressing fibers impairs muscle regrowth after nerve injury. A** Principle component analysis of mRNAseq data generated from GAS muscle of male CON and HSA-MycTG mice 4 (CON, $n = 3$; TG, $n = 4$) or 10 ($n = 4$) days after the first of 5 tamoxifen injections. 4 d and 10 d CON group samples were pooled. **B** Overlap of differentially expressed genes (FDR < 0.05; LFC > 0.5) between CON and either 4 d or 10 d HSA-MycTG groups and **C** Scatterplot of log-fold change and −log10 FDR between CON and 4 d HSA-MycTG for all detected ribosomal protein genes. **D** Pairwise Venn diagram comparisons of genes differentially expressed for acute (3 d AktTG and 4 d HSA-MycTG) and chronic (14 d AktTG and 10 d HSA-MycTG) activation of AKT and overexpression of Myc. Tan (MycTG) and lilac (AktTG) colored arrows indicate direction of change. **E** Gene ontology (Biological process) terms associated with overlapping genes between 14 d AktTG and 10 d HSA-MycTG (**D**, right). **F** Lineage tracing approach to track MuSC-myofiber fusion events following sciatic nerve crush in Pax7-MycTG mice. **G** Representative images and quantification of eGFP+ fiber number (**H**) in TA muscle cross-sections 28 d after nerve crush in Pax7-eGFP

($n = 3$), and Pax7-MycTG ($n = 4$) mice and in contralateral TA muscles from Pax7-MycTG. White arrowhead indicates an eGFP+ MuSC (eGFP image intensity increased compared to fibers). Fiber size distribution of eGFP+ and eGFP- muscle fibers 28 d after nerve crush in TA muscle from (**I**) Pax7-eGFP ($n = 3$) and (**J**) Pax7-MycTG ($n = 4$) mice. **K** Representative image of eGFP+ fibers in longitudinal sections from Pax7-MycTG mice 28 d after nerve crush. **L** Muscle mass 28 days after sciatic nerve crush normalized to the innervated contralateral limb in control (CON, $n = 14$) and Pax7-MycTG ($n = 13$) mice. **M** In vitro specific force in EDL muscles 28 d after nerve crush (and innervated contralateral) in CON (Inn, $n = 13$; 28 d, $n = 14$) and Pax7-MycTG ($n = 13$) mice. **F** Created in BioRender. Ruegg, M. (2025) https://BioRender.com/d74q573. Data are mean ± SEM. Two tailed, independent (**I**, **J**) and paired (**H**) Student's $t$ tests, two-way repeated measures (**L**) or two-way ANOVAs with Sidak's post hoc test (**M**) were used to compare between conditions. For (**B**–**D**), $P$ values were adjusted for multiple comparisons using the Benjamini-Hochberg (FDR) procedure. *, **, and *** denote a significant difference between groups of $P < 0.05$, $P < 0.01$, and $P < 0.001$, respectively.

## Methods

### Animal care

All procedures were performed in accordance with Swiss regulations for animal experimentation and approved by the veterinary commission of the Canton Basel-Stadt. Mice were housed under a fixed 12 h light-dark cycle (6 am to 6 pm) at 22 °C (range 20–24 °C) and 55% (range 45–65%) relative humidity and fed a standard chow diet (KLIBA NAFAG-3432). B6.129S6-Myc$^{tm2Fwa}$/Mmjax mice[29] were purchased from Jax (MMRRC stock #32046) and crossed with HSA-Cre[28], Pax7-Cre$^{ERT2}$[32] or AktTG mice[78] to create HSA-MycKO, Pax7-MycKO and AKT-MycKO mice, respectively. AktTG mice expressing a tamoxifen-inducible constitutively active form of PKBα/Akt1 were obtained from Dr. David Glass at Novartis Institutes for BioMedical Research (NIBR, Cambridge, MA, USA; current address Regeneron Pharmaceuticals, Tarrytown, NY, USA). R26StopFLMYC mice, which contain a CAG-driven human *Myc* downstream of a floxed STOP cassette inserted into the *ROSA26* locus[47] were purchased from Jax (#020458) and crossed with either Pax7$^{CreERT2}$ or HSA$^{MerCreMer}$[48] mice to create tamoxifen inducible Pax7-MycTG and HSA-MycTG mice. For lineage tracing experiments, mice with a Cre-inducible cytosolic CAG-eGFP inserted into the Rosa26 locus[33] were crossed with Pax7-Cre$^{ERT2}$, Pax7-MycKO and Pax7-MycTG mice.

For experiments with HSA-MycKO, HSA-MycTG, Pax7-MycKO and Pax7-MycTG mice, floxed littermates not expressing Cre recombinase were used as controls. For lineage tracing experiments in Pax7-MycKO mice, controls were Pax7-CreERT2 mice homozygous for CAG-eGFP, but not expressing the floxed *Myc* mutation. For lineage tracing experiments in Pax7-MycTG-eGFP mice, homozygous CAG-eGFP mice were crossed with Pax7-MycTG mice and their progeny, all heterozygous for both MycTG and eGFP mutations in the Rosa26 locus, were used. Muscle force measurements showed no difference between mice with 1 or 2 copies of the MycTG mutation and data were pooled for analysis. Age-matched mice of mixed sex, unless specified, were used for each individual experiment. MOV experiments were performed on 3–5-month-old mice, cardiotoxin experiments were performed on 3- to 5-month-old Pax7-MycTG mice and 8- to 11-month-old male Pax7-MycKO mice, nerve crush experiments were performed on 3- to 5-month old Pax7-MycTG and 4- to 10-month old Pax7-MycKO mice, HSA-MycTG experiments were performed on 4- to 5-month old mice, AktTG experiments were performed on 3- to 8-month-old mice. Body composition measurements were performed using an EchoMRI-100 (EchoMRI Medical Systems).

### Experimental models

Mechanical overload was induced in the right *plantaris* (PLA) muscle via surgical ablation of the distal two thirds of its synergist muscles, gastrocnemius and soleus, as previously described[79]. Nerve crush was

performed by exposing and pinching the femoral branch of the sciatic nerve for 10 s with a #55 forceps[79]. For EdU labeling experiments, mice were injected *I.P.* with 100 mg/kg EdU (Invitrogen) 24 h prior to tissue collection. To induce a degeneration/regeneration event, 50 μl of 10 μM cardiotoxin in 0.9% sterile saline was injected into the mid-belly of the TA muscle with a 30 G syringe. In each case, the respective muscle of the contralateral limb served as an internal control. Mice were anaesthetized via isoflurane inhalation. As pain relief, mice received 0.1 mg/kg Buprenorphine (Temgesic) subcutaneously an hour before the procedure and then every 12 h for 48 h.

Tamoxifen (Sigma-Aldrich) diluted in corn oil (Sigma-Aldrich) was administered I.P. at a dose of 75 mg·kg·day$^{-1}$ to Pax7-MycKO and Pax7-MycTG mice, and their respective controls, for 7 days, starting 2 weeks before the experimental intervention, except for CTX experiments in Pax7-MycTG mice, where tamoxifen was administered starting 2 days after CTX injection. For Pax7-CreERT2 and Pax7-MycKO experiments, mice were also administered tamoxifen 3 days per week (Monday, Wednesday, Friday) following surgery to limit any potential expansion of a non-recombined pool of Pax7+ progenitor cells. AktTG, Akt-MycKO mice and their respective CON and MycKO controls were administered tamoxifen (40 mg·kg·day$^{-1}$) I.P. for 4 or 15 consecutive days. Tissue was collected within 6 h of the final tamoxifen injection 3 or 14 days after the first treatment. HSA-MycTG mice and their littermate controls were administered 75 mg·kg·day$^{-1}$ tamoxifen I.P. for 5 consecutive days. Tissue was collected within 6 h of the 5th injection, 4 days after the first injection, or 10 d after the first tamoxifen injection.

### Histology analysis

For lineage tracing experiments with eGFP (Figs. 4B, J and 6N), mice were briefly perfused with PBS and then 4% PFA for 10 min. Muscles were then excised, post-fixed in 4% PFA for 2 h and then dehydrated overnight in 30% sucrose. Fixed or freshly isolated muscles were mounted at resting length in optimal cutting temperature medium (O.C.T, Tissue-Tek) and snap-frozen in thawing isopentane for ~1 min before transfer to liquid nitrogen and storage at −80 °C. Muscle sections (10 μm) were cut from the mid belly at −20 °C on a cryostat (Leica, CM1950), collected on SuperFrost Plus (VWR) adhesion slides and stored at −80 °C. Sections from each experimental condition were always mounted on the same slide to ensure accurate comparisons. For histological analysis, sections were stained with hematoxylin and eosin (H&E; Merck, Zug, Switzerland).

### Immunostaining

Sections were blocked and permeabilized in PBS containing 10% goat serum and 0.4% triton X-100 for 30 min except for Edu labeling experiments, which were fixed in 4% PFA for 10 min and Pax7 and Myogenin stains which were fixed with ice cold methanol for 10 min

and washed 2 × 2 min before being heated to 95 °C for 10 min in 0.01 M sodium citrate buffer. Slides were left to cool and washed once in PBS for 5 min before being blocked for 1 h (or 6 h for Edu labeling) in 0.4% triton X-100, 10% goat serum and mouse on mouse blocking reagent (Vector laboratories, MKB-2213) in PBS.

For fiber typing, sections were incubated for 2 h at RT in a primary antibody solution containing 10% goat serum and SC-71 (IIA, 1:200) or F1.652 (embryonic MHC; 1:100), BF-F3 (IIB, 1:50), BA-D5 (type I, 1:50) which were developed by Prof. Stefano. Schiaffino and obtained from the Developmental Studies Hybridoma Bank developed under the auspices of the National Institute of Child Health and Human Development and maintained by the University of Iowa Department of Biology, and laminin β1γ1 (1:200; #L9393, Sigma). For Pax7 and Myogenin staining, slides were incubated with antibodies against Pax7 (DSHB, 1:20), Myogenin (1:400; Ab124800, Abcam) and laminin β1 (1:100; MA5-14657, Invitrogen). For IgG staining and lineage tracing experiments, sections were incubated overnight at 4 °C in primary antibody solution containing 10% goat serum and antibodies against laminin β1 (1:100; MA5-14657, Invitrogen) or laminin β1γ1 (1:200; #L9393, Sigma), respectively. For Edu labeling experiments, sections were washed with PBST and incubated in primary antibody solution containing 10% goat serum and antibodies against Ki67 (1:200; Ab15580, Abcam) and Pax7 (DSHB, 1:20) overnight at 4 °C.

After incubation in primary antibodies, sections were washed 4 × 10 min in PBS (or 3 × PBST and 1 × PBS for Edu labeling) and then incubated in a secondary antibody solution containing IgG1 GaM Alexa 568 (1:100; #21124, Invitrogen), IgM GaM Alexa 488 (1:100; #21042, Invitrogen), DaRb Alexa647 (1:200; #711-605-152, Jackson) and GaM Dylight 405 (1:50; #115-475-207, Jackson) for fibertyping; Alexa647 (1:200; #21235, Invitrogen) or GaM Cy3 (1:400; #115-165-205, Jackson) for Pax7 and Alexa 488 (1:200; #11034, Invitrogen) for Myogenin and Ki67 staining; GaM IgG Alexa 647 (1:200; #21235, Invitrogen) and GaRt IgG Cy3 (1:500; #112-165-143, Jackson) for IgG staining and; DaRb 647 (1:200; 711-605-152, Jackson) or (Wheat Germ Agglutinin, Alexa 594, 1:200, W-11262, Molecular Probe), for lineage tracing experiments.

After incubation in secondary antibodies, sections were washed 4 × 10 min in PBS and mounted with ProLong™ Gold antifade (Invitrogen). For EdU labeling experiments, prior to mounting, sections were washed 2 × 5 min in PBST and 2 × 5 min in PBS before staining EdU (Alexa 647) using the Click-iT Plus EdU labeling kit, according to the manufacturer's instructions (Invitrogen, C10640). Muscle sections were imaged at the Biozentrum Imaging Core Facility with an Axio Scan.Z1 Slide Scanner (Zeiss) equipped with appropriate band-pass filters. Fiji macros were developed in-house to allow an automated analysis of muscle fiber types (based on intensity thresholds) and muscle cross-sectional area (i.e., minimal Feret's diameter; based on cell segmentation)[80]. All macros and scripts used in this study are available upon request.

## Single fiber isolation and culture

GAS muscles were harvested from both legs and directly placed in 0.2% Collagenase Type I (Sigma) in isolation media (DMEM (Gibco) with 1% Penicillin/Streptomycin) at 37 °C. After 1 h of digestion, the muscles were transferred to warm isolation medium in a dish pre-coated with 20% horse serum (Biological Industries) in isolation medium. The muscles were then flushed using a wide-bore glass pipette to dissociate single fibers. These steps were performed at room temperature for a maximum of 5 min before the dishes were placed at 37 °C for 30 min to allow medium re-equilibration. Once a sufficient number of fibers were obtained, some were fixed immediately for immunostaining while the rest were individually transferred to 24-well dishes containing culture media (20% FBS (PAN-Biotech), 1% Chicken Embryo Extract (Mpbio) and 2.5 ng/mL bFGF (Gibco) in isolation media). These fibers were

incubated at 37 °C in 5% $CO_2$ for 48 h before they were collected and fixed for immunostaining.

## Single fiber staining

For fixation, single fibers were transferred into an Eppendorf pre-coated with 20% horse serum (Biological Industries), washed once with PBS for 5 min, fixed in 4% paraformaldehyde for 10 min and washed three times in 0.1 M glycine in PBS (pH 7.4). Fibers were then permeabilized with 0.2% Triton X-100 in PBS for 10 min at RT before blocking with 10% goat serum in PBS for 3 h at RT. The samples were then incubated with Pax7 (1:20; DSHB), Ki67 (1:200; Ab15580, Abcam) and GFP (1:200; A10262, Thermo Scientific) primary antibodies in 10% goat serum in PBS overnight at 4 °C. They were then washed three times for 5 min in PBS, incubated in GaM Cy3 (1:300; #115-165-205, Jackson) for Pax7, GaRb Cy5 (1:300; #111-175-144, Jackson) for Ki67 and GaC Alexa488 (1:300; #103-545-155, Jackson) for GFP in 10% goat serum in PBS for 1 h at RT. Fibers were then washed three times in PBS and mounted on glass slides with ProLong™ Gold antifade with DAPI (Invitrogen).

## RNA extraction

GAS muscles were pulverized on a metal block cooled with liquid nitrogen before lysis and RNA extraction using the RNeasy Fibrous Tissue Mini Kit (74704, Qiagen). RNA purity and integrity was examined with a Bioanalyser (Agilent).

## Digital PCR

To quantify *Myc* and *Rpl3* expression levels, 450 ng of mRNA from Quad (Fig. 1A) or PLA (Fig. 2C, *Myc* only) muscle were reverse transcribed using the iScript cDNA Synthesis kit (Biorad). Next, dPCR was conducted employing Taqman Gene Expression assays for: *Myc* (Mm00487804_m1; FAM labeled), *Rpl3* (probe-TCTGGAAGCGAC-CATGGCCA; Fwd-CTGAAGTTCATTGACACCACCTC; Rev-GGTCCCA-TAAATGCTTTCTTCTC; Texas Red labeled) and *Hprt* (Mm03024075_m1; VIC labeled) (Life Technologies), utilizing the QIA-cuity One, 5plex Platform system (QIAGEN). The reaction was run in the QIAcuity Nanoplate 26k 24-well plates (QIAGEN). The 40 µl reaction mix consisted of 10 µl of 4x QIAcuity Probe PCR Kit (QIAGEN), 2 µl of the target assay, 2 µl of the reference assay, 23 µl water, and 3 µl of cDNA. Cycling conditions comprised 95 °C for 2 min, followed by 40 cycles of 95 °C for 15 s and 60 °C for 30 s. Absolute quantification of *Myc* and *Rpl3* expression was carried out using the QIAcuity Software Suite (QIAGEN).

## RT-qPCR

RNA purity was determined using a Nanodrop ONEC (Thermo Scientific). cDNA was generated with the iScript™ cDNA Synthesis Kit (Bio-Rad) using 500 ng of extracted RNA according to supplier's manual. cDNA samples were stored at −20 °C. RT-qPCR was performed in duplicate with the LightCycler 480 (Roche Diagnostics) instrument using LightCycler 384-well plates with sealing foil (Roche). The reaction volume of 10 µl contained FastStart Essential DNA Green Master Mix (2X, Roche), forward and reverse primers and cDNA template (1:5 diluted). Primers were designed using Genious®10 software[81] and specificity confirmed by the Basic Local Alignment Search Tool (BLAST)[82]. Potential hairpin formation, complementarity and self-annealing sites were verified to be negative by OligoCalc[83]. The amplification of a single PCR product was confirmed with a melting-point dissociation curve and raw quantification cycle (Cq) values were calculated by a LightCycler 480. Data were analyzed using the comparative Cq method ($2^{-\Delta\Delta Cq}$). Raw Cq values of the target gene (*Myc: Fwd*- GCCAGCCCTGAGCCCCTAGT; *Rev*- GGGTGCGGCGTAGTTGTG CT) were normalized to Cq values of the housekeeping gene (*Gapdh: Fwd*- ACCCAGAAGACTGTGGATGG; *Rev*- GGATGCAGGGATGATGTT CT), which was stable between conditions, and then further normalized to the control group for ease of visualization.

## mRNAseq analysis

RNA concentration was determined with a Quant-iT™ RiboGreen™ RNA assay kit and Qubit flurometer (Invitrogen). Libraries were prepared using the TruSeq stranded mRNA library kit (20020595, Illumina) starting from 200 ng of RNA. Sequencing was performed on the NovaSeq 6000 (Illumina) system (PE 2×51 for Akt-MycKO samples and PE100 for 7-month-old adult and HSA-MycTG samples). FASTQ files were aligned to the indexed mouse transcriptome mm10 using Salmon (version 1.1.0) with the flags validateMappings, seqBias and gcBias. Output quant.sf files from all samples were imported into R (version 4.1.2) using tximeta (Bioconductor) and analyzed with DESeq2 (Bioconductor). Transcript level information was summarized to gene level and genes with fewer than 5 (MycKO and AktTG) or 50 (MycTG) counts across all samples were removed.

## Gene ontology analysis

To annotate genes, we performed gene ontology (GO) analysis using Database for Annotation, Visualization and Integrated Discovery (DAVID)[84]. 'GOTERM_BP_DIRECT', 'GOTERM_MF_DIRECT' and 'GOTERM_CC_DIRECT' categories were used for gene annotation. Background genes for calculating enrichment statistics consisted of all genes expressed in muscle samples. GO terms with an FDR less than 0.05 were considered significantly enriched.

## Western blot analysis

Snap frozen muscles were pulverized on a metal plate chilled in liquid nitrogen. Samples were lysed in ice cold RIPA buffer (50 mM TrisHCl pH 8.0, 150 mM NaCl, 1% NP-40, 0.5% sodium deoxycholate, 0.1% SDS, ddH$_2$O) supplemented with phosphatase and protease inhibitors (Roche), incubated on a rotating wheel for 2 h at 4 °C and sonicated twice for 10 s. Afterwards, the lysate was centrifuged at $16,000 \times g$ for 20 min at 4 °C. Supernatant (cleared lysates) were used to determine total protein amount using the Pierce BCA Protein Assay Kit (Thermo Fisher Scientific) according to the manufacturer's protocol. Proteins were separated on 4–12% Bis-Tris Protein Gels (NuPage Novex, Thermo Fisher Scientific) and transferred to nitrocellulose membrane (GE Healthcare Life Sciences, Amersham). The membrane was blocked with 5% BSA, 0.1% Tween-20, PBS for 1 h at room temperature and then incubated in primary antibody (Myc (D84C12): Rb mAB #5605, Cell Signaling; Rps6 (5G10): Rb mAB, #2217S Cell Signaling; Rps14: 16683-1-AP, Proteintech; or α-actinin: A7732, Ms mAB, Sigma) diluted 1:1000 in blocking solution overnight at 4 °C with continuous shaking. Membranes were then washed with PBS-T (0.1% Tween-20, PBS) for 7 min three times and incubated with secondary horseradish peroxidase-conjugated (HRP) antibody for 1 h at room temperature. After washing with PBS-T, proteins were visualized by chemiluminescence (KPL LumiGLO®, Seracare). Signal was captured on a Fusion Fx machine (VilberLourmat) and analyzed with FUSION Capt FX software.

## RNA fluorescent in situ hybridization (RNA-FISH; RNAscope™)

Slides were fixed in 4% PFA for 1 h at 4 °C and washed 2 × 1 min in PBS before serial dehydration for 5 min in each of 50%, 70% and 2 × 100% ethanol. Slides were stored overnight at −20 °C in 100% ethanol. Slides were then dried for 3 min at RT and circled with a Hydrophobic Barrier Pen (Vector Laboratories, GZ-93951-68) before 30 min protein digestion (protease IV) at RT and washed 3 × 1 min in ddH$_2$O. RNA hybridization with probes against *Myc* (413451), *Pax7* (314181) and either *Cd163* (406631; Fig. 1) or *MyoD1* (316081, Advanced Cell Diagnostics; Fig. 2) and subsequent amplification steps were performed according to the manufacturer's instructions for the Multiplex V2 assay (323100, Advanced Cell Diagnostics) at 40 °C in a HybEZ™ oven (Advanced Cell Diagnostics). TSA-vivid 520 (323271), 570 (323272) and 650 (323273) fluorophores were purchased from Advanced Cell Diagnostics. After hybridization, Slides were blocked 60 min at RT in PBS containing 0.4% Trition X-100 and 3% bovine serum albumin (BSA), washed 2 × 5 min in

PBS and then incubated with primary antibodies against Laminin-β1γ1 (1:200; #L9393, Sigma) in 3% BSA in PBS. Slides were then washed 4 ×10 min in PBS and incubated with Goat anti-Rabbit 750 (A21039, Thermo Fisher) and DAPI in 3% BSA in PBS. Slides were then washed 4 × 10 min in PBS and mounted with ProLong™ Gold antifade (Invitrogen).

## In vitro muscle force

In vitro muscle force was measured in the fast-twitch *extensor digitorum longus* (EDL) and slow-twitch *soleus* muscles. After careful isolation, muscle tendons were tied with surgical suture at each end and mounted on the 1200 A Isolated Muscle System (Aurora Scientific, Aurora, ON, Canada) in an organ bath containing 60 mL of Ringer solution (137 mM NaCl, 24 mM NaHCO$_3$, 11 mM Glucose, 5 mM KCl, 2 mM CaCl$_2$, 1 mM MgSO$_4$, 1 mM NaH$_2$PO$_4$) gassed with 95% O$_2$; 5% CO$_2$ at 30 °C. After defining optimal length, muscles were stimulated with 15 V pulses. Muscle force was recorded in response to 500 ms pulses of 10–250 Hz. Muscle fatigue was assessed by repeated tetanic stimulations at 200 Hz (EDL) or 120 Hz (SOL) every 8 s for 6 min.

## Statistical analysis

All values are expressed as mean ± SEM, unless otherwise stated. Data were tested for normality and homogeneity of variance using a Shapiro–Wilk and Levene's test, respectively. Data were analyzed in GraphPad Prism 8. Student's *t* tests were used for pairwise comparisons. Two-way ANOVAs with Sidak post hoc tests and two-way repeated-measures ANOVAs or mixed-effects models (to account for missing values) for multiple recordings over time, were used to compare between groups with two independent variables. For differential expression analysis of mRNAseq data, *P* values were adjusted for multiple comparisons using the Benjamini-Hochberg (FDR) procedure. Both significant differences ($P < 0.05$) and trends ($P < 0.1$) are reported where appropriate.

## Reporting summary

Further information on research design is available in the Nature Portfolio Reporting Summary linked to this article.

# Data availability

All mRNA-Seq data sets are available at the Gene Expression Omnibus (GEO)[85] under the accession numbers GSE287999 (HSA-MycKO), GSE288001 (HSA-MycTG) and GSE287997 (Akt-MycKO). Source data are provided with this paper.

# Code availability

Code is available upon request.

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

## Acknowledgements

We thank the Biozentrum In-house Imaging Core Facility and the Quantitative Genomics Facility (QGF) at the Department of Biosystems Science and Engineering (D-BSSE, ETH Zürich, Basel) for their technical support. We thank David J. Glass, Chikwendu Ibebunjo and Joseph Cruz (Novartis Institutes for Biomedical Research, Cambridge, USA) for providing us with Akt-TG mice. This work was supported by the Cantons of Basel-Stadt and Basel-Landschaft and a grant from the Swiss National Science Foundation (220244) to M.A.R. and D.J.H..

## Author contributions

D.J.H. conceived the project, secured funding, designed and performed experiments, analyzed data, prepared figures and wrote the manuscript. S.L. performed muscle force measurements, immunostaining and mouse perfusions. F.O. performed intraperitoneal injections, genotyping and western blotting. M.S. and B.M.B. performed experiments, analyzed data and contributed to writing of the manuscript. E.M. performed digital qPCR experiments. T.J.M. performed single fiber experiments. M.A.R. secured funding, supervised the project and wrote the manuscript.

## Competing interests

The authors declare no competing interests.
