## [Transparent Peer Review file · Nature Communications]

Muscle fiber Myc is dispensable for muscle growth and its forced expression severely perturbs homeostasis

Corresponding Author: Professor Markus Ruegg

Version 0:

Reviewer comments:

Reviewer #1

(Remarks to the Author)

This manuscript by Ham et al, describes the surprising dispensability for the proto-oncogene Myc in post-natal muscle growth and repair. The authors show through elegant and well-controlled genetic experiments that the expression of Myc in the skeletal myofiber is dispensable for growth. In addition, deletion of Myc did not affect Akt-induced skeletal muscle growth. Also of surprise in this study was the effect that upon overexpression, Myc was deleterious to muscle homeostasis rather than the anticipatory enhancement. Overall, this is an exceptionally well-conducted study from which the conclusions are well-supported by the presented data. However, there are two major weakness. The first is that much of the data that supports the dispensability of Myc for skeletal muscle growth was generated in 7-month old mice which are considered young adults. At this stage, post-natal skeletal muscle growth has markedly subsided. Therefore, it is conceivable that the lack of effect could simply be a result of the much slower growth rate in mice at this age. Although the authors examined younger mice, these were female – again without significant effect. This reviewer would suggest performing similar experiments in 2-month old male mice as more supportive evidence of there being a lack of effect of Myc on young and old post-natal muscle growth. As indicated the data is of exceptional quality however, the data albeit surprising lack significant impact in that it does not implicate and/or even suggest why Myc exhibits this anomalous dispensability for muscle growth. The reviewers perform a series of RNA-seq “omic” experiments but do not provide outside of a general description of gene changes how and why Myc is dispensable. This point should at least be discussed.

Comments:

1. Mice study might require examination in younger mice, to ensure that the lack of effect of Myc is not simply related to the age of the mice.
2. Could the authors provide some insight in the Discussion as to why Myc exhibits such a deleterious effect when overexpressed.
3. The authors conclude that loss of Myc does not impair the ability of constitutively active Akt to promote muscle growth. This is not particularly surprising since the actions of Akt on muscle growth are largely independent of the actions of Myc. Indeed, the authors themselves demonstrate this when they perform their RNAseq analysis in Akt constitutively active and MycKO mice. Therefore, it is unclear to this reviewer as to what the value of these experiments. The authors should re-evaluate the relevance of these studies or at least provide a more compelling rationale for their inclusion.
4. The effects of Myc on muscle stem cell proliferation as determined in these studies is somewhat ambiguous. The authors appear to measure eGFP+ fibers and subsequently assign this as a measure of myofiber growth. Although a good proxy for proliferation it is possible that the effects are not through proliferation but rather hypertrophy. Therefore, the authors need to directly assess the effects of Myc on MuSC proliferation by performing in vivo Brdu-labeling to directly assess this effect.
5. Can the dispensability of Myc, which is presumably the c-Myc isoform, be subserved by one or more of the other Myc isoforms such as the p64/67 isoforms. The authors should comment on this possibility.

Reviewer #2

(Remarks to the Author)

Myc is a well-known oncogenic transcriptional factor and a potent inducer of ribosomal protein genes involved in protein synthesis. Myc was believed to be an essential transcription factor for muscle hypertrophy based on various lines of evidence which are described in introduction:

Contrary to this theory, this study shows that Myc is not required for surgical muscle hypertrophy, recovery from atrophy due to sciatic nerve crush, or Akt-dependent hypertrophy. While, MuSC-specific Myc deficiency impairs hypertrophy, and abnormal MuSC dynamics were observed in MycKO, even during recovery from atrophy caused by sciatic nerve crush. Forced expression of Myc in myofibers did not induce muscle hypertrophy and had a negative effect on myofibers. Overall, the data are convincing and sufficient to overturn previous theories. Please consider the following comments:

Specific comments

1) Could L-Myc and N-Myc, other than c-Myc, compensate? Is c-Myc dominantly expressed within the Myc family in skeletal muscle?

2) Figure 2B

In this experiment, results of males and females were mixed. Please state the number of males and females to indicate the extent to which male and female bias among the samples may have affected the results. Additionally, please add information on the sexes in Figures 3A-D, 6H, K, and 7L.

3) Figure 3E-J and Discussion (lines 404-406)

In this experiment, there is no direct evidence showing the normal proliferation ability of cKO MuSCs. Because MuSC behaviors in overloaded muscle may differ from those in regenerating muscles, the authors need to examine the number of MuSCs to support their conclusions. Additionally, if fusion is abnormal, a reduction in the number of myonuclei should be seen in MycKO MOV-28d compared to control. This reviewer recommends presenting data showing what happens in MycKO-MuSC in this model.

In males, the weight of PLA was reduced after MOV compared with that of contralateral muscle. Additionally, the response of myofiber type also differs between males and females. Is there a possibility that male muscle is more susceptible to muscle damage caused by MOV compared to females?

5) Figure 4

This experiment is positioned as an experimental system in which the growth of MuSCs hardly occurs and fusion is observed. However, could it be that the MuSCs are proliferating slowly? If only fusion is occurring, then MuSCs should be exhausted.

6) Lines 244; Next we quantified the number of eGFP+ and eGFP- centrally nucleated fibers, which is a proxy for recent fusion events.

This is not case for murine models. Recent Kardon's group mentioned this issue. Dev Cell. 2024 Mar 29;S1534-5807(24)00184-9.

7) Lines 248; These findings indicate that skeletal muscle can tolerate a lack of MuSC proliferation/fusion during muscle regrowth after nerve injury, perhaps by relying more heavily on fusion-competent, progenitor pools not expressing Pax7. Indeed, bone marrow-derived hematopoietic stem cells, among others, have been shown to be able to fuse to fibers and support the regeneration of injured muscle [36]. Therefore, contributions from such cells could explain the full recovery of muscle size in Pax7-MycKO mice after nerve crush.

It is recommended that this description be changed, as a Pax7-negative, Integrin $\alpha 7$ + SMMC seems more likely than a BM story.

Mol Cell. 2019 May 2;74(3):609-621.e6.

8) Is the construct described in Fig. 6A and 6E correct? Alternatively, reviewers speculate the CAG promoter induce Myc transcripts by removing the stop codon.

Reviewer #3

(Remarks to the Author)

In this manuscript "Muscle fiber Myc is dispensable for muscle growth and forced expression severely perturbs homeostasis", Ham et al. characterized the KO and OE effects of c-myc in both skeletal muscle and muscle stem cells. The authors conducted the study to examine the role of Myc in muscle hypertrophy and muscle stem cell homeostasis. They found that when Myc was deleted from skeletal muscle using the human skeletal actin Cre, the KO muscle can still undergo hypertrophy in response to mechanical overload and PKB/Akt-activation. When Myc was deleted from muscle stem cells, the KO mice failed to respond to mechanical overload induced muscle hypertrophy. On the other hand, when Myc was overexpressed in skeletal muscle, it was detrimental for the muscle. When Myc was overexpressed in muscle stem cells, regeneration process was hindered because the newly formed myofibers could not mature. These animal models elucidate the distinct roles of Myc in muscle stem cells and skeletal muscle, however, there are many inconsistencies (texts and figures) in the paper that needs to be taken care of. Some writings are confusing.

Major concerns:

Line 75-89. The authors showed that there was only 60% reduction of myc mRNA in a muscle specific KO model driven by

HSACre, besides, the myc downstream targets were not altered. How can the authors conclude this KO model is a successful one? The staining in Figure 1B showed very minimal foci of Myc even in the HSA-MycKO, how do the authors explain for the remaining 40% Myc mRNA? Apart from the mRNA data, please also include western blots for Myc protein in CON and HSA-MycKO muscle tissues. A more efficient KI Cre driver, such as Myl1Cre, should be used to, at least, confirm if the phenotype is similar.

Figure 1B: HSA-MycKO inset #1. The authors were trying to show that the non-muscle cells contribute to the high abundance of myc mRNA after the KO, yet the CD163+ cell (macrophages) not have myc staining? Please clarify what is the main cellular source of myc expression in the muscle. Figure 1D, Line 93-94. The graph shows an increase of EDL/SOL/Gas muscle weight at 5-7 months, what was the interpretation for this result? Figure 1E, Line 95-96. The author concluded in the manuscript that there was no difference in the specific force of EDL or SOL muscle. However, the graph clearly demonstrated a significant increase of SOL specific force. Please consider to better describe/interpret the data and results throughout the manuscript.

Line 121-125: The authors performed synergistic ablation to induce PLA muscle hypertrophy in CON and MycKO mice to examine the requirement of Myc for MOV. However, first of all, since the KO was not successful or the level of residual Myc protein might be substantial, how can the authors conclude Myc is dispensable? Second, how do the authors compare their results showing there is not a difference in the muscle weight to other published studies showing a 20-30% increase in the PLA muscle mass after the surgery at day 7?

Figure 2C: There is an obvious outlier in the MOV-7d CON group which brings the average of the group high. Please consider adding more repeats to confirm the changes. Besides, there are only 3 repeats in the HSA-MycKO group, the variation in the group is also very high. In order to make meaningful conclusions, please consider increasing the statistical power. Figure 2B: The MOV-3d in MycKO group is missing in the graph. Please add the data in the graph and manuscript accordingly. Figure 2H: It is very difficult to directly compare the size of the myofiber in different groups (CON vs. MycKO). Please remake the graph so that the CON-sham and MycKO-sham are directly compared and CON-MOV and MycKO-MOV are directly compared.

Figure 2H and 3G: In these two graphs, there is a discrepancy in the IIX fiber size in the CON group. How do the author explain the larger IIA and IIX myofiber size in 2H yet the no change of IIA and smaller IIX fiber size in 3G in the same CON group?

Line 211-231: The author showed by eGFP labeling that there was almost no GFP+ myofibers in the Pax7MycKO mice. Since it was shown that Myc inhibits myoblasts differentiation, and hence Myc deletion in Pax7 promotes differentiation (PMID: 29786076). How do the authors explain the lack of MuSC contribution to myofiber hypertrophy when they are prone to differentiate.

Line 246-253: The authors think that the relative similar number of centra-nuclei in ctrl and Pax7MycKO mice was due to the contribution of progenitor cell pools not expressing Pax7 and citing PMID: 15733662, however, the reference only showed a 0.4% contribution of non-muscle stem cell derived myofibers from the parabiosis model. This conclusion must be supported by experiments.

Line 293-297, Figure 5I & 5J: The authors stated that "gene ontology terms were largely enriched and depleted to similar extents in CON and HSA-MycKO mice after 14d PKB /Akt activation, with the exception of terms related to ribosome biogenesis, which were specifically depleted in AktTG mice.". First of all, what does "specifically depleted in AktTG mice" mean. Second, based on the figure, the AKT-MycKO mice have fewer terms related to ribosome biogenesis compared to AKT-TG mice. Doesn't this suggest that there could be a AKT signaling defect after Myc deletion?

Line 380-384: These sentences are very difficult to understand. The sentence to show that there was no eGFP+ fibers in the control muscle was unnecessary. Figure 7H, please add the contralateral leg in the graph.

Minor concerns:

The title is missing a term after "forced expression" (forced expression of xxx).

Line 17 "we tested the hypothesis...". The authors used the word hypothesis, yet there is no hypothesis that is brought forward. The previous sentences summarized the findings from PMID: 34423683, yet what is the actual hypothesis of this manuscript? Please elaborate.

Line 27: "by changing the size, number and composition of the fibers...". Here the authors stated that the skeletal muscle myofibers can increase the number, please cite proper citations that showed the increase of myofiber numbers as the current paradigm is that the number of fibers do not change after p21.

Line 52: "...muscle growth are induced in transgenic mice.". It is not clear which transgenic mice the authors are referring to. There are issues of improper referencing throughout the manuscript.

Line 99: "indicating a strong reduction...". Based on the evidence shown by the authors, they cannot conclude this is a strong reduction.

Line 94-95: What is the age for the mice for the force frequency experiment?

Line 99: "...5-7 months of age". What is the rationale for choosing 5-7months of age for the RNA seq experiments?

Line 112: "...relatively slow accumulation of mass during postnatal muscle development...". How do the authors define "a relatively slow mass accumulation"? There is not a significant change in the body mass, or muscle weight (except for higher EDL/SOL/Gas weight) in the MycKO mice.

Line 160-161: The authors conclude that muscle fiber Myc is not necessary for MOV induced muscle hypertrophy. This may be due the low KO efficiency. The results also contradict the notion that there is a significant contribution of MuSCs to the nuclei and cellular content during MOV. As shown by the authors later, the KO of Muscle stem cell Myc led to a blunted hypertrophy response after MOV. In this case, the authors cannot make such conclusion as the true effect of Myc in muscle fiber may be masked.

Line 215: "...the proliferative burden on MuSCs is mild". Please add references to this statement. Besides, in line 375, the authors wrote "we took advantage of the mild MuSC fusion observed during the regrowth phase...". Please explain which process is mild during this denervation induced regrowth period, whether it is proliferation or fusion? Or both?

Line 258: The authors wrote "investigate the possibility that muscle fiber Myc expression is necessary to support myofiber hypertrophy". Didn't the authors already "conclude" that myofiber Myc is dispensable for muscle hypertrophy?

Line 285-291: The comparisons presented here are very confusing and difficult to understand. Please rewrite the sentences.

Line 302-304: since there is already a published study that injected AAV6 to mediate the overexpression of Myc in skeletal muscle (PMID: 34423683). What is the rationale of the authors for these set of experiments?

Reviewer #4

(Remarks to the Author)

Version 1:

Reviewer comments:

Reviewer #1

(Remarks to the Author)

The authorsd have thoroughly addressed the concerns of this reviewer and no further experimentation or modification is requested.

Reviewer #2

(Remarks to the Author)

The authors sincerely addressed all concerns raised by this reviewer. This paper will be a conclusive study showing the role of Myc in muscle hypertrophy and homeostasis.

Reviewer #3

(Remarks to the Author)

The authors did a great job addressing my comments.

Reviewer #4

(Remarks to the Author)

Response to reviewer comments

We thank all reviewers for their thorough and thoughtful review of our manuscript. We have performed additional experiments, added data and improved text to address all reviewers' concerns and believe these additions greatly strengthened the manuscript.

REVIEWER COMMENTS

Reviewer #1 (Remarks to the Author):

This manuscript by Ham et al, describes the surprising dispensability for the proto-oncogene Myc in post-natal muscle growth and repair. The authors show through elegant and well-controlled genetic experiments that the expression of Myc in the skeletal myofiber is dispensable for growth. In addition, deletion of Myc did not affect Akt-induced skeletal muscle growth. Also of surprise in this study was the effect that upon overexpression, Myc was deleterious to muscle homeostasis rather than the anticipatory enhancement. Overall, this is an exceptionally well-conducted study from which the conclusions are well-supported by the presented data. However, there are two major weakness. The first is that much of the data that supports the dispensability of Myc for skeletal muscle growth was generated in 7-month old mice which are considered young adults. At this stage, post-natal skeletal muscle growth has markedly subsided. Therefore, it is conceivable that the lack of effect could simply be a result of the much slower growth rate in mice at this age. Although the authors examined younger mice, these were female – again without significant effect. This reviewer would suggest performing similar experiments in 2-month old male mice as more supportive evidence of there being a lack of effect of Myc on young and old post-natal muscle growth. As indicated the data is of exceptional quality however, the data albeit surprising lack significant impact in that it does not implicate and/or even suggest why Myc exhibits this anomalous dispensability for muscle growth. The reviewers perform a series of RNA-seq “omic” experiments but do not provide outside of a general description of gene changes how and why Myc is dispensable. This point should at least be discussed.

Comments:

1.1 Mice study might require examination in younger mice, to ensure that the lack of effect of Myc is not simply related to the age of the mice.

We thank the reviewer for this important point. We have now addressed the possibility of muscle fiber Myc expression playing a role in muscle development within the more rapid period of post-natal growth by reporting body mass (Fig. 1D) and EDL muscle force on 2-month-old and 5- to 7-month-old male CON and HSA-MycKO mice (Fig. 1F-G). To investigate the effect of muscle fiber Myc depletion on accumulation of post-natal muscle mass, we compared the relationship between muscle mass and body mass across a range of ages for males (left) and females (right) for TA, TRI, GAS and Quad muscles (Fig. 1E), and report TA, QUAD, EDL, SOL, PLA, GAS and TRI mass for 2-month-old males (Fig. S1A) and 5- to 7-month-old males (Fig. S1B) and females (Fig. S1C). These new data and comparisons further confirm that Myc depletion does not negatively affect post-natal muscle development.

1.2. Could the authors provide some insight in the Discussion as to why Myc exhibits such a deleterious effect when overexpressed.

We thank the reviewer for raising this important point. While requiring future experimental interrogation, we believe the deleterious effect of Myc results from Myc promoting a 'dedifferentiation' signature, reminiscent of that seen in terminally differentiated epidermal cells

during skin wound healing. Whether dedifferentiation is possible in skeletal muscle fibers remains unclear, but would presumably be catastrophic for the structure and function of the muscle fibers themselves. Thus, the strong decrease in expression of genes encoding for key structural and contractile proteins (e.g. Myh4, Myl1 and Acta1) in HSA-MycTG mice likely explains the sharp drop in muscle function and integrity. We have added context and details for this point both in the results and discussion sections. Text is as follows:

Results:

Line 419-426: *“These juxtaposed phenotypic outcomes coincided with expression of muscle differentiation and maturation markers. For example, 3d of PKB α /Akt1 activation promoted the expression of Myod1 and Mef2c, key transcription factors promoting muscle differentiation [52], along with mature muscle markers such as Myh4, Myl1 and Acta1. On the other hand, 4d of MycTG expression strongly reduced the expression of the muscle differentiation transcription factor Mef2c as well as Maf [53], a transcription factor promoting maturation of fast-type skeletal muscle, along with many genes encoding for key muscle (particularly fast-type) proteins, including Myh1, Myh4, Tnnc2, Tnni2, Mylpf, Myl1, and Tnnt3.”*

Discussion

Line 560-568: *“Perhaps indicating a Myc-driven reprogramming towards a less differentiated state, genes encoding the mature muscle transcription factors Mef2c and Maf along with key muscle structural proteins, including Myh1, Tnnc2, Tnni2, Mylpf, Myl1, Tpm1, Tnnt3, Acta1 and Myh4 were strongly suppressed in MycTG muscle just 96 hours after the first tamoxifen injection. The widespread downregulation of mature muscle markers is suggestive of a ‘dedifferentiation-like’ response to Myc overexpression in muscle fibers. Myc-mediated dedifferentiation has previously been described in terminally differentiated epidermal cells during skin wound healing [77]. Not surprisingly, the outcome of shifting transcriptional focus away from characteristic mature muscle genes led to rapid disintegration of muscle fiber morphology and loss of muscle force.”*

1.3. The authors conclude that loss of Myc does not impair the ability of constitutively active Akt to promote muscle growth. This is not particularly surprising since the actions of Akt on muscle growth are largely independent of the actions of Myc. Indeed, the authors themselves demonstrate this when they perform their RNAseq analysis in Akt constitutively active and MycKO mice. Therefore, it is unclear to this reviewer as to what the value of these experiments. The authors should re-evaluate the relevance of these studies or at least provide a more compelling rationale for their inclusion.

We agree with the reviewer that our rationale for these experiments needed to be strengthened and the wording of the discussion gave the wrong impression that it is already known that Akt-mediated muscle fiber hypertrophy is independent of Myc. As the reviewer points out, the experiments in Fig. 6 now clearly demonstrate that AKT-mediated muscle fiber hypertrophy is in fact independent of Myc; however, this was never previously tested in skeletal muscle and there is clear evidence that the AKT-mTORC1 pathway and Myc interact and cooperate to control cell growth (although these studies are largely performed in cancer cells). For example, inhibition of Myc strongly blunts the growth of AKT/RAS-driven liver cancer (HCC), while Myc overexpression enhances AKT-driven liver cancer growth (Xin, *Oncogene*, 2017). Furthermore, previous theories of how Myc may contribute to muscle growth, propose complementary interactions of Myc and the Akt-mTORC1 pathway in mediating growth (Chaillou, *J Cell Physiology*, 2014). We have now strengthened the rationale for performing these experiments in the results section, and expanded our interpretation of the results within the discussion. See specific text below.

Results:

Line 293-300: *“As MOV-induced muscle growth was heavily MuSC-dependent, we next investigated the possibility that muscle fiber Myc expression is necessary to support an increase in muscle fiber size, i.e. hypertrophy. Activation of the PKB/Akt-mTORC1 pathway in muscle fibers promotes robust*

hypertrophy [40]. In non-muscle cells, this pathway and Myc promote cell growth through intertwined and often co-dependent mechanisms, with Myc-driven cell growth depending on mTORC1 activity [41] and Akt-driven growth responsive to Myc activity [42]. Myc and the Akt-mTORC1 pathway have also been hypothesized to function cooperatively in skeletal muscle hypertrophy [43], but this has not been directly tested.”

Discussion:

Line 497-510: “While the Akt-mTORC1 pathway and Myc activate overlapping cell growth pathways, they also cooperate and the activity of both have been shown to be obligatory for tissue/tumor growth mediated by proliferative cells. Indeed, Akt-mTORC1 signaling increases Myc protein abundance through enhanced translation efficiency [62] and protein stabilization [63] and inhibiting Myc strongly blunts tumor growth [42]. Similarly, Myc-driven tumor growth relies on mTORC1 signaling [41]. MuSC-mediated myogenesis is also severely impaired by loss of either Akt-mTORC1 signaling [64] or Myc (Fig. 3-5). The similar gene expression signatures observed in MycTG and AktTG mice (Fig. 8A-E) indicate that Akt-mTORC1 and Myc signaling also overlap in terminally differentiated skeletal muscle fibers. In contrast, we provide the first evidence that Myc is not required for robust PKB/Akt-mediated skeletal muscle fiber hypertrophy, suggesting alternate, Myc-independent pro-growth processes in skeletal muscle fibers. PKBa/Akt1 activation expands ribosomal protein content translationally via mTORC1 [65, 66], while ribosomal RNA synthesis can be mediated via casein kinase II alpha (Csnk2a1) and transcription initiation factor I (TIF1A), encoded by the Trim24 gene [67], perhaps obviating the need for Myc activity in muscle fiber hypertrophy.”

1.4. The effects of Myc on muscle stem cell proliferation as determined in these studies is somewhat ambiguous. The authors appear to measure eGFP+ fibers and subsequently assign this as a measure of myofiber growth. Although a good proxy for proliferation it is possible that the effects are not through proliferation but rather hypertrophy. Therefore, the authors need to directly assess the effects of Myc on MuSC proliferation by performing in vivo Brdu-labeling to directly assess this effect.

We agree that eGFP-based lineage tracing of Pax7+ MuSCs only serves as a gross measure of MuSC function. While the role of Myc on muscle stem cells has been previously reported (e.g. Ref: 26 and 27), we agree with the comments from all three reviewers (point 1.4, 2.3, 2.5 and 3.10) that the effect of Pax7-MycKO on MuSC proliferation was inadequately addressed in the manuscript. We have now completed additional experiments in Pax7-MycKO mice to directly assess the impact of Myc depletion on MuSC activation and proliferation ex vivo using single fiber preparations (Fig. 3) and in vivo using EdU labeling in the nerve crush model of muscle regrowth (Fig. 5). We think these experiments thoroughly address the reviewer’s concerns and strongly improve the manuscript. Please see below for specific text.

Results

Line 181-195: “As Myc is known to play a central role in cell proliferation, including in muscle progenitor cells [27], we compared proliferation competence in Pax7-eGFP and Pax7-MycKO MuSCs by isolating single fibers from mice that were injected for 5 consecutive days with tamoxifen (Fig. 3A). Single fibers were either fixed immediately (0h) or cultured in growth medium for 48 h (48h) and then stained for Pax7 and Ki67, a marker of cell cycle activation (Fig. 3A-B). Even though the isolation process can disturb MuSC quiescence, just one third of eGFP+ cells on fibers from Pax7-eGFP mice were Ki67+ at 0h, while 48 hours of culture in growth medium increased the percentage of Ki67+, eGFP+ cells to 94% (Fig. 3C). Similarly, 92% of eGFP+ cells from Pax7-eGFP mice at 0h existed as isolated cells on muscle fibers, while 48 hours of culture significantly increased the percentage of eGFP+ cells in clusters of 2, 3-4 and 5+ cells at 48h (Fig. 3B, D), indicating strong MuSC proliferation. In contrast, Myc depletion severely blunted MuSC proliferation, with just 38% of eGFP+ cells expressing Ki67 and >80% existing as isolated cells on muscle fibers from Pax7-MycKO mice cultured for 48 hours in growth media.”

Line 270-277: *“As Myc mRNA expression peaked 14 days after nerve crush (Fig. 5F), we examined the effect of Myc depletion on regrowth-induced MuSC proliferation by injecting mice with EdU 24 h before collecting tissue 14 d after nerve crush, thereby labeling all cells that replicated their DNA within this period. Nerve crush significantly increased (+54%) the number of Pax7+ MuSCs in whole GAS muscle cross-sections compared to contralateral limbs in CON mice, but not Pax7-MycKO mice (Fig. 5G-H). Similarly, the number of Ki67+ (Fig. 5G, I) and EdU+ (Fig. 5J) MuSCs were strongly elevated 14d after nerve crush in CON but not Pax7-MycKO mice.”*

Discussion

Line 455-457: *“Using Pax7-MycKO mice, we confirmed that Myc is vital for MuSCs to mount an appropriate myogenic response to regenerative or growth pressure [26], concomitant with strong impairments in MuSC activation and proliferation ex vivo (Fig. 3) and in vivo (Fig. 5).”*

1.5. Can the dispensability of Myc, which is presumably the c-Myc isoform, be subserved by one or more of the other Myc isoforms such as the p64/67 isoforms. The authors should comment on this possibility.

We thank the reviewer for this important point. For Myc floxed mice (Moreno de Alboran et al., 2011), the floxed sites encompass both exon 2 and 3 of c-Myc, thereby targeting all c-Myc transcript isoforms. Please also see Reviewer comment 2.1 regarding potential compensatory upregulation of the Myc paralogs Mycn and Mycl.

Reviewer #2 (Remarks to the Author):

Myc is a well-known oncogenic transcriptional factor and a potent inducer of ribosomal protein genes involved in protein synthesis. Myc was believed to be an essential transcription factor for muscle hypertrophy based on various lines of evidence which are described in introduction: Contrary to this theory, this study shows that Myc is not required for surgical muscle hypertrophy, recovery from atrophy due to sciatic nerve crush, or Akt-dependent hypertrophy. While, MuSC-specific Myc deficiency impairs hypertrophy, and abnormal MuSC dynamics were observed in MycKO, even during recovery from atrophy caused by sciatic nerve crush. Forced expression of Myc in myofibers did not induce muscle hypertrophy and had a negative effect on myofibers. Overall, the data are convincing and sufficient to overturn previous theories.

Please consider the following comments:

Specific comments

2.1. Could L-Myc and N-Myc, other than c-Myc, compensate? Is c-Myc dominantly expressed within the Myc family in skeletal muscle?

We thank the reviewer for raising an important point. We have now added data (Fig. S1C) showing that Mycn and Mycl are lowly expressed in adult skeletal muscle and HSA-MycKO does not influence expression. The lack of compensatory upregulation might be the result of the different chromosomal location of c-Myc, n-Myc and l-Myc. Along with the added data, we provide a detailed explanation of this point in the discussion. Please see below for specific text.

Results

Line 120-122: *“Importantly, the strong reduction in Myc expression did not coincide with compensatory upregulation of the other Myc-family paralogs Mycl and Mycn, which were expressed at levels ~10-fold lower than Myc in 5- to 7-month-old CON mice (Fig. S1F).”*

Discussion

Line 524-537: *“A possible explanation for the dispensability of Myc in skeletal muscle fibers could be compensatory upregulation of the Myc family paralogs Mycn (N-Myc) or Mycl (L-Myc), which share high sequence similarity with Myc (i.e. C-Myc) and have been shown to functionally compensate for loss of Myc, when appropriately expressed [69]. However, Myc, Mycn and Mycl are located on different chromosomes (1, 2 and 8, respectively), allowing divergent transcriptional regulation and therefore distinct tissue and temporal expression patterns [70]. Indeed, whole-body knockout of either Myc or Mycn are lethal between embryonic day 9 and 11 [71, 72]; however, when the Myc coding sequence is replaced by the corresponding Mycn coding sequence, mice are born, reach adulthood and are able to reproduce [69]. Interestingly, skeletal muscle development was strongly impaired in mice expressing Mycn instead of Myc, indicating a specific reliance on Myc for skeletal muscle development. In line with distinct regulation of Myc-family expression, Mycn and Mycl expression was ~10-fold lower than Myc expression in skeletal muscle lysates from CON mice with no sign of compensatory upregulation in HSA-MycKO mice.”*

2.2. Figure 2B. In this experiment, results of males and females were mixed. Please state the number of males and females to indicate the extent to which male and female bias among the samples may have affected the results. Additionally, please add information on the sexes in Figures 3A-D, 6H, K, and 7L.

We have now added female and male numbers in the relevant figures.

2.3. Figure 3E-J and Discussion (lines 404-406). In this experiment, there is no direct evidence showing the normal proliferation ability of cKO MuSCs. Because MuSC behaviors in overloaded muscle may differ from those in regenerating muscles, the authors need to examine the number of MuSCs to support their conclusions. Additionally, if fusion is abnormal, a reduction in the number of myonuclei should be seen in MycKO MOV-28d compared to control. This reviewer recommends presenting data showing what happens in MycKO-MuSC in this model.

This was also suggested by reviewer 1 and 3. While we no longer have a license to perform synergist ablation surgery, we have now performed EdU labeling experiments in the nerve crush model of muscle regrowth to (at least partly) characterize the impact of Myc depletion on MuSC activation and proliferation in muscle growth. In addition, we have also addressed whether Myc depletion in MuSCs affects their proliferation using single fibers. Please see our response to point 1.4 of reviewer#1 for more detail.

2.4. In males, the weight of PLA was reduced after MOV compared with that of contralateral muscle. Additionally, the response of myofiber type also differs between males and females. Is there a possibility that male muscle is more susceptible to muscle damage caused by MOV compared to females?

The reviewer raises an interesting point. There is certainly growing recognition of sex-specific responses to a wide variety of interventions. Although the Myc depletion in MuSCs strongly impaired MOV-induced muscle growth in both males and females, we thought it was important to report these subtle differences to aid in continuing efforts to understand sex-specific differences in muscle biology. Indeed, a greater susceptibility to muscle damage could explain this difference. An alternate hypothesis is that the MOV-induced stimulus is greater in males due to their larger mass and loss of larger and stronger synergist muscles (GAS and SOL). While precisely pinpointing the underlying mechanism could be challenging and is beyond the scope of this study, we think the question is worthy of future investigation.

2.5. Figure 4. This experiment is positioned as an experimental system in which the growth of MuSCs hardly occurs and fusion is observed. However, could it be that the MuSCs are proliferating slowly? If only fusion is occurring, then MuSCs should be exhausted.

We agree with the reviewer that this point needed more clarification. We have now measured MuSC population size, activation status and proliferation (EdU) at 14d post nerve crush, where muscle regrowth commences (please see more detailed description in response to reviewer point 1.4). At this point, MuSC activation and expansion of the MuSC population is blunted in Pax7-MyckKO mice. Lineage tracing experiments complement this new data by showing that incorporation of eGFP+ MuSCs into muscle fibers is strongly reduced in Pax7-MyckKO mice 28 and 42d post nerve crush. See Fig. 5G.

2.6. Lines 244; Next we quantified the number of eGFP+ and eGFP- centrally nucleated fibers, which is a proxy for recent fusion events.

This is not case for murine models. Recent Kardon's group mentioned this issue. Dev Cell. 2024 Mar 29:S1534-5807(24)00184-9.

We thank the reviewer for pointing this out. We have reworded the sentence accordingly. Please see below.

Line 278-279: *"Next we quantified the number of eGFP+ and eGFP- centrally nucleated fibers, which result from muscle fiber regenerative events involving progenitor cell fusion [20]."*

2.7. Lines 248; These findings indicate that skeletal muscle can tolerate a lack of MuSC proliferation/fusion during muscle regrowth after nerve injury, perhaps by relying more heavily on fusion-competent, progenitor pools not expressing Pax7. Indeed, bone marrow-derived hematopoietic stem cells, among others, have been shown to be able to fuse to fibers and support the regeneration of injured muscle [36]. Therefore, contributions from such cells could explain the full recovery of muscle size in Pax7-MyckKO mice after nerve crush.

It is recommended that this description be changed, as a Pax7-negative, Integrin α 7+ SMMC seems more likely than a BM story.

We agree that Pax7-negative SMMCs would be a more likely non-MuSC candidate. We have updated the statement and referenced Giordani et al., Mol Cell, 2019. See below for text:

Line 284-287: *"Indeed, a population of Pax7-negative cells with myogenic potential has been identified within skeletal muscle, expressing markers of smooth muscle and mesenchymal cells (SMMCs) [38]. Furthermore, bone marrow-derived hematopoietic stem cells have also been shown to fuse to fibers and support the regeneration of injured muscle [39]."*

2.8. Is the construct described in Fig. 6A and 6E correct? Alternatively, reviewers speculate the CAG promoter induce Myc transcripts by removing the stop codon.

We thank the reviewer for identifying the error in our schematic. We have now updated 7A and 7E so that CAG precedes the Lox-STOP-Lox Myc insertion.

Reviewer #3 (Remarks to the Author):

In this manuscript "Muscle fiber Myc is dispensable for muscle growth and forced expression

severely perturbs homeostasis”, Ham et al. characterized the KO and OE effects of c-myc in both skeletal muscle and muscle stem cells. The authors conducted the study to examine the role of Myc in muscle hypertrophy and muscle stem cell homeostasis. They found that when Myc was deleted from skeletal muscle using the human skeletal actin Cre, the KO muscle can still undergo hypertrophy in response to mechanical overload and PKB/Akt-activation. When Myc was deleted from muscle stem cells, the KO mice failed to respond to mechanical overload induced muscle hypertrophy. On the other hand, when Myc was overexpressed in skeletal muscle, it was detrimental for the muscle. When Myc was overexpressed in muscle stem cells, regeneration process was hindered because the newly formed myofibers could not mature. These animal models elucidate the distinct roles of Myc in muscle stem cells and skeletal muscle, however, there are many inconsistencies (texts and figures) in the paper that needs to be taken care of. Some writings are confusing.

Major concerns:

3.1. Line 75-89. The authors showed that there was only 60% reduction of myc mRNA in a muscle specific KO model driven by HSA-Cre, besides, the myc downstream targets were not altered. How can the authors conclude this KO model is a successful one? The staining in Figure 1B showed very minimal foci of Myc even in the HSA-MycKO, how do the authors explain for the remaining 40% Myc mRNA? Apart from the mRNA data, please also include western blots for Myc protein in CON and HSA-MycKO muscle tissues. A more efficient KI Cre driver, such as Myl1Cre, should be used to, at least, confirm if the phenotype is similar.

We agree that a lack of phenotype always triggers the question of whether this is due to a failure to actually delete the gene of interest. There are several lines of correlative evidence that HSA-Cre mice are very efficient in deleting a gene. This mouse model has been used by us to delete floxed genes (e.g. *Rptor* and *Tsc1* – e.g. Bentzinger, 2013 [REF 80]; Castets, 2013 [PMID: 23602450]) and others (e.g. Schwander et al., 2003 [REF 59]). In all those experiments, mRNA from muscle lysates was still present, which is likely due to the fact that only ~half of all nuclei within muscle reside within muscle fibers (i.e. myonuclei – See Fig. R1 below). Hence, the ~60% reduction in Myc mRNA in whole-muscle lysates is rather high. It is also clear from experiments in Pax7-MycKO mice that Myc depletion is highly efficient when combined with Pax7-Cre. Therefore, it is also very likely that the floxed *Myc* gene is deleted by the expression of Cre recombinase. These two lines of evidence make us confident that Myc depletion in HSA-MycKO mice was successful. We are unaware of direct comparisons between the efficiency of HSA and Myl1 promoters for Cre expression, however, Myl1-Cre has been shown to limit Cre expression to fast-type fibers (Mourkioti, Genesis, 2008; PMID: 18693277), while HSA-Cre is expressed in all fiber types.

Despite the strong correlative evidence above, we use smRNA-FISH (Fig. 2D-E) to distinguish the cellular localization of *Myc* expression in CON and HSA-MycKO mice in response to MOV (which stimulates *Myc* expression). In CON muscle, we counted an average of 60 myonuclei expressing Myc RNA per plantaris cross-section 7d after synergist ablation but only 3 myonuclei per section expressing Myc RNA in HSA-MycKO mice, indicating ~95% efficiency of muscle fiber Myc depletion. Of the few Myc+ myonuclei in HSA-MycKO mice, the majority were associated with newly regenerating muscle fibers. Unfortunately, we have not been able to detect measurable levels of Myc in muscle lysates of either wild-type or HSA-MycKO mice using Western blot analysis because of the low expression.

We hope the reviewer agrees that our combined data strongly support high Myc depletion efficiency in both Pax7- and HSA-MycKO mice.

3.2. Figure 1B: HSA-MycKO inset #1. The authors were trying to show that the non-muscle cells contribute to the high abundance of myc mRNA after the KO, yet the CD163+ cell (macrophages) not have myc staining? Please clarify what is the main cellular source of myc expression in the muscle.

The reviewer is correct that our goal was to show that *Myc* expression remains in *Pax7+* MuSCs and non-muscle fiber cells, using *Cd163+* macrophages as an example. While *Cd163* staining did provide an example of mRNA expression patterns in non-muscle fiber cells, we indeed did not find strong *Myc* staining in *Cd163+* cells and agree that this created confusion. We have now removed *Cd163* staining from Fig 1B to improve clarity (see amended text below). With regards to the main cellular source of *Myc*, publically available snRNAseq data sets (e.g. MyoAtlas), as well as our own (see table R1; doi.org/10.1101/2024.05.15.594276), suggest that *Myc* is occasionally expressed in most cell types. In our own data, we generated transcriptomes from ~36k nuclei, 55% of which were muscle fiber nuclei. Overall, we detected *Myc* transcripts in approximately 0.6% of the myonuclei and 0.6% of non-muscle fiber nuclei, suggesting that *Myc* is expressed at low levels in most cell types.

	Cell type	#Cells	# Cells expressing	
			Myc	pctExpress
Myo-nuclei	Type IIB	14950	99	0.66
	Type IIX	3071	9	0.29
	Type IIA	844	2	0.24
	MTJ	622	4	0.64
	NMJ	272	0	0.00
Non-muscle fiber cells	FAPs	9821	68	0.69
	Immune cells	2202	16	0.73
	Endothelial	1944	7	0.36
	MuSCs	554	5	0.90
	Mmrrn1+	221	3	1.36
	Adipocytes	404	3	0.74
	Smooth muscle	199	2	1.01
	Schwann cells	94	1	1.06
	Tenocytes	231	1	0.43
	Pericytes	563	0	0.00
	Peri- and Endoneurial	199	0	0.00

Table R1: *Myc* expression in myonuclei and non-muscle fiber nuclei in snRNA-seq data generated from 2xTA and 2xGAS muscles (Ham A., et al., BioRxiv, 2024).

Results

Line 83-88: *“To distinguish Myc mRNA expression originating from muscle fiber nuclei, we co-stained sections with antibodies against laminin $\beta 1\gamma 1$ (to outline muscle fibers) and included Pax7 probes to identify MuSCs, which are located between the basement membrane and sarcolemma and could easily be confused for muscle fiber nuclei. Myc mRNA could be clearly visualized in a small population of Pax7-expressing MuSCs, along with other mononucleated cells residing between muscle fibers, in muscle sections from both CON and HSA-MycKO mice (Fig. 1B).”*

3.3. Figure 1D, Line 93-94. The graph shows an increase of EDL/SOL/Gas muscle weight at 5-7 months, what was the interpretation for this result?

Indeed, there is a small elevation in muscle mass for some muscles. However, this was not consistent across muscles or experiments. After adding a sizeable 2-month-old male control group (n=8-12) and splitting the 5-7-month-old muscle mass data into males (n=5-6) and females (5-7), we no longer see

significant differences in muscle mass between genotypes (Fig. S1A-C). However, when combining data for all ages, we can still observe differences between genotypes in the properties of linear relationships between body mass and muscle mass with slightly higher mass of some muscles in MycKO than CON mice (new Fig. 1E). While this data may indicate that even basal levels of muscle fiber Myc expression is detrimental to muscle growth, as we see in HSA-MycTG mice, due to the minor difference and inconsistency across muscles and experiments, we decided to be conservative by concluding only that Myc is not necessary for normal muscle development, which is solidly supported by the data.

3.4. Figure 1E, Line 95-96. The author concluded in the manuscript that there was no difference in the specific force of EDL or SOL muscle. However, the graph clearly demonstrated a significant increase of SOL specific force. Please consider to better describe/interpret the data and results throughout the manuscript.

We thank the reviewer for pointing out this inconsistency. We have re-examined the statistical analysis for this graph and realized that the reported difference was the result of a graph copy-paste issue. Using a 2-WAY repeated measures ANOVA with Fisher's LSD post-hoc tests, there were no significant differences between genotypes for any stimulation frequency or fatigue stimulation (see new Fig. S1E).

3.5. Line 121-125: The authors performed synergistic ablation to induce PLA muscle hypertrophy in CON and MycKO mice to examine the requirement of Myc for MOV. However, first of all, since the KO was not successful or the level of residual Myc protein might be substantial, how can the authors conclude Myc is dispensable?

Please see detailed response to the question of HSA-MycKO efficiency in 3.1

3.6. Second, how do the authors compare their results showing there is not a difference in the muscle weight to other published studies showing a 20-30% increase in the PLA muscle mass after the surgery at day 7?

This is indeed a valid point and we agree that the time line of muscle growth in our ablation model is slightly delayed compared to most other reported data. This could of course be an experimental difference and we have therefore "calibrated" our model by initial time course studies to identify time points where substantial muscle growth could be observed and we still see more than a 60% increase in muscle mass within 4 weeks. All the experiments were conducted by Dr. Shuo Lin, a senior research associate who has conducted such experiments for many years, and surgeries on CON and MycKO mice were always performed in the same experiment. It is important to note that published work shows that there is a wide variation in the extent and time frame of muscle growth and the contribution of MuSCs in this model. We would also like to emphasize that our manuscript includes multiple additional experimental models (i.e. AKT-induced hypertrophy for HSA-MycKO and nerve crush-induced re-growth for Pax7-MycKO). In all of these experimental models the conclusion concerning Myc's role in muscle growth were consistent, thereby compensating for the potential weaknesses of the synergist ablation model.

3.7. Figure 2C: There is an obvious outlier in the MOV-7d CON group which brings the average of the group high. Please consider adding more repeats to confirm the changes. Besides, there are only 3 repeats in the HSA-MycKO group, the variation in the group is also very high. In order to make meaningful conclusions, please consider increasing the statistical power.

Figure 2B: The MOV-3d in MycKO group is missing in the graph. Please add the data in the graph and manuscript accordingly.

We appreciate your comment, and we agree that the response of muscle mass to synergist ablation surgery can be variable, likely reflecting the extent of inflammation. After an initial time course study

including CON mice at 3d, 7d, 14d and 28d and MycKO mice at 7d, 14d and 28d we suspected that: 1) changes in muscle mass at 3d were largely related to inflammation/edema, 2) strong muscle growth was not yet evident at 7d, and 3) robust muscle growth was most evident at the 28d time point. For this reason and because synergist ablation surgery represents a substantial burden for the mouse, we chose to increase mouse numbers for the 28d time point only. We agree that for symmetry it would have been ideal to have included a 3d MycKO group and increase the numbers for the 7d time point; however, given variation at 3 d and the strong inflammation, and that, if anything, muscle mass tended to be higher at 7d in MycKO mice than CON mice, we believe the current data sufficiently answers our primary question of whether muscle fiber Myc is necessary to support MOV-induced muscle growth. As we are aware of these shortcomings, we have been careful not to make strong claims about any potential differences in mass between CON and MycKO groups. Of note, we no longer have an animal license to perform synergist ablation surgery and hence we cannot perform additional MOV experiments.

3.8. Figure 2H: It is very difficult to directly compare the size of the myofiber in different groups (CON vs. MycKO). Please remake the graph so that the CON-sham and MycKO-sham are directly compared and CON-MOV and MycKO-MOV are directly compared.

We have now combined CON and HSA-MycKO data in the same graph. There are no statistical differences between genotypes for any fiber type or condition.

3.9. Figure 2H and 3G: In these two graphs, there is a discrepancy in the IIX fiber size in the CON group. How do the author explain the larger IIA and IIX myofiber size in 2H yet the no change of IIA and smaller IIX fiber size in 3G in the same CON group?

The reviewer is correct that the response to ablation surgery was slightly different for experiments with HSA-MycKO (Fig. 2) and Pax7-MycKO (previously Fig. 3, now Fig. 4) mice. In both cases, plantaris muscle mass strongly increases along with IIA and IIX fiber **number**. We suspect that the difference in fiber size between experiments represents a slightly different time course in the muscles response to MOV. For example, in Fig. 2, newly created fibers might be further along in their growth process than those presented in Fig. 4. Although the same person performed the surgery for both experiments, they were performed ~12 months apart and in different mouse lines, which could potentially explain the difference. In both experiments though, CON and MycKO mice underwent surgery at the same time and hence the comparisons within the experiment are of course valid.

3.10. Line 211-231: The author showed by eGFP labeling that there was almost no GFP+ myofibers in the Pax7MycKO mice. Since it was shown that Myc inhibits myoblasts differentiation, and hence Myc deletion in Pax7 promotes differentiation (PMID: 29786076). How do the authors explain the lack of MuSC contribution to myofiber hypertrophy when they are prone to differentiate.

We agree that high, full-length Myc levels inhibit differentiation. It is important to note that Myc can be cleaved via Calpains (as is the case in muscle progenitor cells) producing Myc-Nick (Ref: 27; PMID: 2069196), which promotes differentiation. In our case, the lack of Myc strongly impairs MuSC activation and proliferation, impairing expansion of the MuSC pool and limiting the number of cells that can fuse with fibers. Additionally, Myc depletion also means Myc-Nick depletion, which may also impair differentiation. We now provide more compelling evidence for the proliferation deficit in MycKO MuSCs in Fig 3A-D and Fig 5G-J (please see more detailed description in response to reviewer point 1.4).

3.11. Line 246-253: The authors think that the relative similar number of central-nuclei in ctrl and Pax7MycKO mice was due to the contribution of progenitor cell pools not expressing Pax7 and citing PMID: 15733662, however, the reference only showed a 0.4% contribution of non-muscle stem cell derived myofibers from the parabiosis model. This conclusion must be supported by experiments.

We agree that there are more likely candidates for non-MuSC progenitor cell contributions to muscle fibers. E.g. Pax7-negative SMMCs as suggested by Reviewer 2. We also emphasize that this is merely a plausible explanation rather than a conclusion, and directly testing this idea would be outside the scope of this study. We have updated the statement and referenced Giordani et al., Mol Cell, 2019. See updated text in response to point 2.7

3.12. Line 293-297, Figure 5I & 5J: The authors stated that “gene ontology terms were largely enriched and depleted to similar extents in CON and HSA-MycKO mice after 14d PKB α /Akt activation, with the exception of terms related to ribosome biogenesis, which were specifically depleted in AktTG mice”. First of all, what does “specifically depleted in AktTG mice” mean. Second, based on the figure, the AKT-MycKO mice have fewer terms related to ribosome biogenesis compared to AKT-TG mice. Doesn't this suggest that there could be a AKT signaling defect after Myc deletion?

In accordance with the reviewer's comment, we have simplified and expanded the text describing these data. In addition, we have included additional graphs (Fig S5A-C) of ribosomal protein gene expression, which show that the same patterns in ribosome-related gene expression exist in both Akt-TG and Akt-MycKO groups; however, the magnitude of the changes are higher for Akt-TG than Akt-MycKO. The slightly lower (but non-significant) expression of ribosomal protein genes in MycKO mice (see Fig 1K) could be responsible for impacting the fold change from Akt-MycKO to MycKO. See below for specific text:

Results:

Line 336-345: *“In addition to the comparable magnitude of PKB α /Akt1-induced gene expression changes between CON and HSA-MycKO mice, gene ontology terms (Biological process, Molecular function and Cellular component) were largely enriched (Fig. 6I) and depleted (Fig. 6J) to similar extents in CON and HSA-MycKO mice after 14d PKB α /Akt1 activation, with the exception of terms related to ribosome biogenesis, which were more strongly depleted in AktTG than Akt-MycKO mice. Further examination of ribosomal protein gene expression showed strong downregulation in both 14d AktTG and Akt-MycKO mice (Fig. S5A-B), compared to CON and MycKO mice, respectively. The fold change in ribosomal protein gene expression was also strongly correlated between these two comparisons, but the slope of the relationship indicated greater downregulation of gene expression in 14d AktTG than Akt-MycKO mice (Fig. S5C). Together, these results indicate that Myc depletion in muscle fibers does not impair PKB α /Akt1-induced muscle hypertrophy or gene expression changes.”*

3.13. Line 380-384: These sentences are very difficult to understand. The sentence to show that there was no eGFP+ fibers in the control muscle was unnecessary. Figure 7H, please add the contralateral leg in the graph.

We have added the contralateral leg for Fig 7H (Now Fig. 8H) and simplified the text as follows:

Line 433-435: *“28 days after nerve crush, TA muscle cross sections from Pax7-eGFP-MycTG and Pax7-eGFP mice displayed a comparable elevation in the number of eGFP+ fibers (Fig. 8G-H).”*

Minor concerns:

3.14. The title is missing a term after “forced expression” (forced expression of xxx).

To make it clearer that we refer to Myc, without repeating the word Myc, we have updated the title to “Muscle fiber Myc is dispensable for muscle growth and its forced expression severely perturbs homeostasis”. We asked several people and they all thought that the title is now clear.

3.15. Line 17 “we tested the hypothesis...”. The authors used the word hypothesis, yet there is no hypothesis that is brought forward. The previous sentences summarized the findings from PMID: 34423683, yet what is the actual hypothesis of this manuscript? Please elaborate.

We have clarified the statements within the Abstract as follows:

Line 15-19: *“Furthermore, muscle fiber Myc overexpression recapitulates many aspects of growth-related gene expression, leading to the hypothesis that Myc mediates pro-growth responses to anabolic stimuli, such as exercise. Here, we tested this hypothesis by examining mouse models in which Myc was specifically eliminated or overexpressed in skeletal muscle fibers or muscle stem cells (MuSC).”*

3.16. Line 27: “by changing the size, number and composition of the fibers...”. Here the authors stated that the skeletal muscle myofibers can increase the number, please cite proper citations that showed the increase of myofiber numbers as the current paradigm is that the number of fibers do not change after p21.

We agree that under normal circumstances fiber number is largely unaltered in adulthood. However, it is clear that in response to an appropriate stimulus (i.e. synergist ablation), the muscle is capable of increasing the number of muscle fibers either by fiber splitting or hyperplasia. See Jorgenson et al., 2020 (REF 31) and Fig. 2F-H & 4C-D.

3.17. Line 52: “...muscle growth are induced in transgenic mice.”. It is not clear which transgenic mice the authors are referring to. There are issues of improper referencing throughout the manuscript.

We have now updated the text to specifically refer to ‘Myc transgenic mice’.

Line 52-54: *“However, although gene expression signatures consistent with muscle growth are induced in Myc transgenic mice, the necessity of muscle fiber Myc expression for skeletal muscle growth has not been directly tested.”*

3.18. Line 99: “indicating a strong reduction...”. Based on the evidence shown by the authors, they cannot conclude this is a strong reduction.

We could not find the specific sentence the reviewer is referring to – perhaps line 90 “Despite the strong reductions in muscle fiber Myc expression in HSA-MycKO mice, there was...”. Based on the detailed response to point 3.1 regarding the effectiveness of Myc depletion in HSA-MycKO mice, we think this remains an accurate statement.

3.19. Line 94-95: What is the age for the mice for the force frequency experiment?

Specific details for the age of the mice in these experiments has now been placed on the graph (See Fig. 1F-G and SA-E)

3.20. Line 99: “...5-7 months of age”. What is the rationale for choosing 5-7months of age for the RNA seq experiments?

We chose this time point partly because of convenience (i.e. they were available for the experiment) and because they represent adult mice, where muscle growth has finished. In any case, we have now strengthened the conclusions for this part by including a 2-month-old male group (as per the recommendation of reviewer 1).

3.21. Line 112: “...relatively slow accumulation of mass during postnatal muscle development...”. How do the authors define “a relatively slow mass accumulation”? There is not a significant change in

the body mass, or muscle weight (except for higher EDL/SOL/Gas weight) in the MycKO mice.

We agree that the definition of what constitutes postnatal growth rate (e.g. rate of absolute mass accumulation or % increase in mass) will change the interpretation of whether this is slow or fast. Since the goal of this sentence was to point out a potential weakness of the experiment and suggest that perhaps we have not sufficiently stressed muscle growth processes (that may or may not require Myc), we have altered the sentence to remove statements about growth rates during development. See below:

Line 122-126: "Together, these data indicate that muscle fiber Myc expression is not necessary for normal post-natal muscle growth, despite a downregulation of genes associated with muscle structure and development, potentially indicating that Myc-regulated growth processes are not sufficiently stressed during development for growth to be limited by its absence."

3.22. Line 160-161: The authors conclude that muscle fiber Myc is not necessary for MOV induced muscle hypertrophy. This may be due the low KO efficiency. The results also contradict the notion that there is a significant contribution of MuSCs to the nuclei and cellular content during MOV. As shown by the authors later, the KO of Muscle stem cell Myc led to a blunted hypertrophy response after MOV. In this case, the authors cannot make such conclusion as the true effect of Myc in muscle fiber may be masked.

Regarding the question of KO efficiency, please refer to our response to point 3.1. We agree though that the contribution of MuSCs during MOV-induced muscle growth could compensate for a lack of muscle-fiber Myc. To overcome this limitation, we generated Akt-MycKO mice where it has been shown that their hypertrophic response is independent of MuSCs (Blaauw et al., 2009 [REF 24]) - see also the response to 3.24. below.

3.23. Line 215: "...the proliferative burden on MuSCs is mild". Please add references to this statement. Besides, in line 375, the authors wrote "we took advantage of the mild MuSC fusion observed during the regrowth phase...". Please explain which process is mild during this denervation induced regrowth period, whether it is proliferation or fusion? Or both?

We agree that this statement was somewhat of an assumption based on eGFP+ fibers. We have now measured the number of MuSCs (Pax7+), as well as their level of activation (i.e. Ki67) and proliferation 'rates' (i.e. EdU incorporation) 14d after nerve crush. Compared to cardiotoxin, which results in a 10-fold increase in MuSC number 4 days post injury (Fig. 3G), Nerve crush causes a 50% increase in MuSC numbers at 14d (Fig. 5H), ~12% of which are Ki67+ (Fig 5I). Furthermore, EdU incorporation indicates ~7% of cells remaining Pax7+ (note: this number does not include cells that proliferate and differentiate during this 24h period) incorporated EdU over the preceding 24 h period (Fig 5J). We also changed the word 'mild' to 'lower'.

3.24. Line 258: The authors wrote "investigate the possibility that muscle fiber Myc expression is necessary to support myofiber hypertrophy". Didn't the authors already "conclude" that myofiber Myc is dispensable for muscle hypertrophy?

While we agree that 'Muscle hypertrophy' is generally used interchangeably with 'Muscle growth' and often used to describe an increase in muscle mass, muscle growth could result from an increase in myofiber size (i.e. myofiber or muscle fiber hypertrophy) or an increase in muscle fiber number (i.e. hyperplasia) or both. As muscle growth in response to synergist ablation can result from both myofiber hypertrophy and hyperplasia, we wanted to specifically test Myc's necessity for myofiber hypertrophy, which can be potently stimulated by Akt activation. We have reviewed our use of nomenclature throughout and now use 'Muscle growth' to refer to a general increase in muscle

mass, while myofiber or muscle fiber hypertrophy refers to an increase in the cross sectional area of the muscle fibers themselves.

3.25. Line 285-291: The comparisons presented here are very confusing and difficult to understand. Please rewrite the sentences.

We have simplified this text. It now reads as follows:

Line 329-333: *“Compared to CON mice, PKB α /Akt1 activation differentially regulated (Adjusted $P < 0.05$; Log fold change > 0.5) 2418 genes after 3d and 4221 genes after 14d, with 3369 genes differentially regulated between 3d and 14d (Fig. 6H). In Akt-MycKO mice, PKB α /Akt1 activation differentially regulated a comparable number of genes after 3d (2224) and 14d (4405) compared to MycKO mice, while 3017 genes were differentially regulated between 3 and 14d.”*

3.26. Line 302-304: since there is already a published study that injected AAV6 to mediate the overexpression of Myc in skeletal muscle (PMID: 34423683). What is the rationale of the authors for these set of experiments?

The study by Mori et al., uses a CMV promoter to express Myc, which will cause expression of Myc in any cell type infected by AAV6 within the muscle, whereas the HSA-Cre promoter is highly specific for skeletal muscle fibers. Secondly, Mori et al., did not examine potential changes in muscle structure or function resulting from Myc overexpression. Our data confirm previous studies (including Mori et al.) with regard to Myc-induced signaling, but add important context regarding the resulting outcome of these signaling changes on skeletal muscle homeostasis.

Reviewer #4 (Remarks to the Author):
